# COMPOSITIONAL TRANSFER IN HIERARCHICAL REINFORCEMENT LEARNING

## ABSTRACT

The successful application of flexible, general learning algorithms to real-world robotics applications is often limited by their poor data-efficiency. To address the challenge, domains with more than one dominant task of interest encourage the sharing of information across tasks to limit required experiment time. To this end, we investigate compositional inductive biases in the form of hierarchical policies as a mechanism for knowledge transfer across tasks in reinforcement learning (RL). We demonstrate that this type of hierarchy enables positive transfer while mitigating negative interference. Furthermore, we demonstrate the benefits of additional incentives to efficiently decompose task solutions. Our experiments show that these incentives are naturally given in multitask learning and can be easily introduced for single objectives. We design an RL algorithm that enables stable and fast learning of hierarchical policies and the effective reuse of both behavior components and transition data across tasks in an off-policy setting for complex, real-world domains. Finally, we evaluate our algorithm in simulated environments as well as physical robot experiments and demonstrate substantial improvements in data data-efficiency over competitive baselines.

## 1 INTRODUCTION

While recent successes in deep (reinforcement) learning for computer games (Atari (Mnih et al., 2013), StarCraft (Vinyals et al., 2019)), Go (Silver et al., 2017) and other high-throughput domains, e.g. (OpenAI et al., 2018), have demonstrated the potential of these methods in the big data regime, the high cost of data acquisition has so far limited progress in many tasks of real-world relevance. Data efficiency in machine learning generally relies on inductive biases to guide and accelerate the learning process; e.g. by including expert domain knowledge of varying granularity. Incorporating such knowledge can accelerate learning – but when inaccurate it can also inappropriately bias the space of solutions and lead to sub-optimal results.

Robotics represents a domain in which data efficiency is critical, and human prior knowledge is commonly provided. However, for scalability and reduced dependency on human accuracy, we can instead utilise an agent's permanent embodiment and shared environment across tasks. Intuitively, such a scenario suggests the natural strategy of focusing on inductive biases that facilitate the sharing and reuse of experience and knowledge across tasks while other aspects of the domain can be learned. As a general principle this relieves us from the need to inject detailed knowledge about the domain, instead we can focus on general principles that facilitate reuse (Caruana, 1997).

Successes for transfer learning have, for example, built on optimizing initial parameters (e.g. Finn et al., 2017), sharing models and parameters across tasks either in the form of policies or value functions (e.g. Rusu et al., 2016; Teh et al., 2017; Galashov et al., 2018), data-sharing across tasks (e.g. Riedmiller et al., 2018; Andrychowicz et al., 2017), or through the use of task-related auxiliary objectives (Jaderberg et al., 2016; Wulfmeier et al., 2017). Transfer between tasks can, however, lead to either constructive or destructive transfer for humans (Singley and Anderson, 1989) as well as for machines (Pan and Yang, 2010; Torrey and Shavlik, 2010). That is, jointly learning to solve different tasks can provide both benefits and disadvantages for individual tasks, depending on their similarity. Finding a mechanism that enables transfer where possible but avoids interference is one of the long-standing research challenges.

In this paper we explore the benefits and limitations of hierarchical policies in single and multitask reinforcement learning. Similar to Mixture Density Networks (Bishop, 1994) our models represent policies as state-conditional Gaussian mixture distributions, with separate Gaussian mixture components as low-level policies which can be selected by the high-level controller via a categorical action choice. In the multitask setting, to obtain more robust and versatile low-level behaviors, we additionally shield the mixture components from information about the task at hand. In this case, task information is only communicated through the choice of mixture component by the high-level controller, and the mixture components can be seen as domain-dependant, task-independent skills although the nature of these skills is not predefined and emerges during end-to-end training.

We implement this idea by building on three forms of transfer: targeted exploration via the concatenation of tasks within one episode (Riedmiller et al., 2018), sharing transition data across tasks (Andrychowicz et al., 2017; Riedmiller et al., 2018), and reusing low-level components of the aforementioned policy class. To this end we develop a novel robust and data-efficient multitask actor-critic algorithm, Regularized Hierarchical Policy Optimization (RHPO). Our algorithm uses the multitask learning aspects of SAC (Riedmiller et al., 2018) to improve data-efficiency and robust policy optimization properties of MPO (Abdolmaleki et al., 2018a) in order to optimize hierarchical policies. We furthermore demonstrate the generality of hierarchical policies for multitask learning via improving results also after replacing MPO as policy optimizer with another gradient-based, entropy-regularized policy optimizer (Heess et al., 2015) (see Appendix A.10).

We demonstrate that compositional, hierarchical policies – while strongly reducing training time in multitask domains – can fail to improve performance in single task domains if no additional inductive biases are given. While multitask domains provide sufficient pressure for component specialization, and the related possibility for composition, we are required to introduce additional incentives to encourage similar developments for single task domains. In the multitask setting, we demonstrate considerably improved performance, robustness and learning speed compared to competitive continuous control baselines demonstrating the relevance of hierarchy for data-efficiency and transfer. We finally evaluate our approach on a physical robot for robotic manipulation tasks where RHPO leads to a significant speed up in training, enabling it to solve challenging stacking tasks on a single robot [1].

## 2 PRELIMINARIES

We consider a multitask reinforcement learning setting with an agent operating in a Markov Decision Process (MDP) consisting of the state space $\mathcal{S}$, the action space $\mathcal{A}$, the transition probability $p(s_{t+1}|s_t, a_t)$ of reaching state $s_{t+1}$ from state $s_t$ when executing action $a_t$ at the previous time step $t$. The actions are drawn from a probability distribution over actions $\pi(a|s)$ referred to as the agent's policy. Jointly, the transition dynamics and policy induce the marginal state visitation distribution $p(s)$. Finally, the discount factor $\gamma$ together with the reward $r(s, a)$ gives rise to the expected reward, or value, of starting in state $s$ (and following $\pi$ thereafter) $V^\pi(s) = \mathbb{E}_\pi[\sum_{t=0}^\infty \gamma^t r(s_t, a_t)|s_0 = s, a_t \sim \pi(\cdot|s_t), s_{t+1} \sim p(\cdot|s_t, a_t)]$. Furthermore, we define multitask learning over a set of tasks $i \in I$ with common agent embodiment as follows. We assume shared state, action spaces and shared transition dynamics across tasks; tasks only differ in their reward function $r_i(s, a)$. Furthermore, we consider task conditional policies $\pi(a|s, i)$. The overall objective is defined as

$$J(\pi) = \mathbb{E}_{i \sim I} \left[ \mathbb{E}_{\pi, p(s_0)} \left[ \sum_{t=0}^\infty \gamma^t r_i(s_t, a_t) | s_{t+1} \sim p(\cdot|s_t, a_t) \right] \right] = \mathbb{E}_{i \sim I} \left[ \mathbb{E}_{\pi, p(s)} \left[ Q^\pi(s, a, i) \right] \right],$$

(1)

where all actions are drawn according to the policy $\pi$, that is, $a_t \sim \pi(\cdot|s_t, i)$ and we used the common definition of the state-action value function – here conditioned on the task – $Q^\pi(s, a, i) = \mathbb{E}_\pi \left[ \sum_{t=0}^\infty \gamma^t r_i(s_t, a_t) | a_0 = a, s_0 = s, a_t \sim \pi(\cdot|s_t, i), s_{t+1} \sim p(\cdot|s_t, a_t) \right]$.

---

[1]Additional videos for task and component visualization are provided under https://sites.google.com/view/rhpo/

## 3 METHOD

This section introduces Regularized Hierarchical Policy Optimization (RHPO) which focuses on efficient training of modular policies by sharing data across tasks; extending the data-sharing and scheduling mechanisms from Scheduled Auxiliary Control with randomized scheduling (SAC-U) (Riedmiller et al., 2018). We start by introducing the considered class of policies, followed by the required combination – and extension – of MPO (Abdolmaleki et al., 2018a) and SAC-U (Riedmiller et al., 2018) for training structured hierarchical policies in a multitask, off-policy setting.

### 3.1 HIERARCHICAL POLICIES

We start by defining the hierarchical policy class which supports sharing sub-policies across tasks. Formally, we decompose the per-task policy $\pi(a|s,i)$ as

$$\pi_\theta(a|s,i) = \sum_{o=1}^{M} \pi_\theta^L\left(a|s,o\right) \pi_\theta^H\left(o|s,i\right), \tag{2}$$

with $\pi^H$ and $\pi^L$ respectively representing a "high-level" switching controller (a categorical distribution) and a "low-level" sub-policy (components of the resulting mixture distribution), where $o$ is the index of the sub-policy. Here, $\theta$ denotes the parameters of both $\pi^H$ and $\pi^L$, which we will seek to optimize. While the number of components has to be decided externally, the method is robust with respect to this parameter (Appendix A.8.3).

Note that, in the above formulation *only the high-level controller $\pi_H$ is conditioned on the task information $i$*; i.e. we employ a form of information asymmetry (Galashov et al., 2018; Tirumala et al., 2019; Heess et al., 2016) to enable the low-level policies to acquire general, task-independent behaviours. This choice strengthens decomposition of tasks across domains and inhibits degenerate cases of bypassing the high-level controller. Intuitively, these sub-policies can be understood as building reflex-like low-level control loops, which perform domain-dependent but task-independent behaviours and can be modulated by higher cognitive functions with knowledge of the task at hand.

### 3.2 DATA-EFFICIENT MULTITASK POLICY OPTIMIZATION

In the following sections, we present the equations underlying RHPO. For the complete pseudocode algorithm the reader is referred to the Appendix A.2.1. To optimize the policy class described above we build on the MPO algorithm (Abdolmaleki et al., 2018a) which decouples the policy improvement step (optimizing $J$ independently of the policy structure) from the fitting of the hierarchical policy. Concretely, we first introduce an intermediate non-parametric policy $q(a|s,i)$ and consider optimizing $J(q)$ while staying close, in expectation, to a reference policy $\pi_{ref}(a|s,i)$

$$\max_q J(q) = \mathbb{E}_{i \sim I}\left[\mathbb{E}_{q, s \sim \mathcal{D}}\left[\hat{Q}(s,a,i)\right]\right], \text{s.t. } \mathbb{E}_{s \sim \mathcal{D}, i \sim I}\quad\left[\text{KL}\left(q(\cdot|s,i)\|\pi_{ref}(\cdot|s,i)\right)\right] \leq \epsilon, \tag{3}$$

where $\text{KL}(\cdot\|\cdot)$ denotes the Kullback Leibler divergence, $\epsilon$ defines a bound on the KL, $\mathcal{D}$ denotes the data contained in a replay buffer, and assuming that we have an approximation of the ground-truth state-action value function $\hat{Q}(s,a,i) \approx Q^\pi(s,a,i)$ available (see Equation (4) for details on learning $\hat{Q}$ from off-policy data). Starting from an initial policy $\pi_{\theta_0}$ we can then iterate the following steps to improve the policy $\pi_{\theta_k}$:

**Policy Evaluation:** Update $\hat{Q}$ such that $\hat{Q}(s,a,i) \approx \hat{Q}^{\pi_{\theta_k}}(s,a,i)$, see Equation (4).

**Policy Improvement:**

- **Step 1:** Obtain $q_k = \arg\max_q J(q)$, under KL constraints with $\pi_{ref} = \pi_{\theta_k}$ (Equation (3)).
- **Step 2:** Obtain $\theta_{k+1} = \arg\min_\theta \mathbb{E}_{s \sim \mathcal{D}, i \sim I}\left[\text{KL}\left(q_k(\cdot|s,i)\|\pi_\theta(\cdot|s,i)\right)\right]$, under additional regularization (Equation (6)).

**Multitask Policy Evaluation** For data-efficient off-policy learning of $\hat{Q}$ we build on scheduled auxiliary control with uniform scheduling (SAC-U) (Riedmiller et al., 2018) which exploits two main

ideas to obtain data-efficiency: i) experience sharing across tasks; ii) switching between tasks within one episode for improved exploration.

Formally, we assume access to a replay buffer containing data gathered from all tasks, which is filled asynchronously to the optimization (similar to e.g. Espeholt et al. (2018)) where for each trajectory snippet $\tau = \{(s_0, a_0, R_0), \ldots, (s_L, a_L, R_L)\}$ we record the rewards for all tasks $R_t = [r_{i_1}(s_t, a_t), \ldots, r_{i_{|I|}}(s_t, a_t)]$ as a vector in the buffer. Using this data we define the retrace objective for learning $\hat{Q}$, parameterized via $\phi$, following (Munos et al., 2016; Riedmiller et al., 2018) as

$$\min_{\phi} L(\phi) = \sum_{i \sim I} \mathbb{E}_{\tau \sim \mathcal{D}} \Big[ \big( r_i(s_t, a_t) + \gamma Q^{ret}(s_{t+1}, a_{t+1}, i) - \hat{Q}_{\phi}(s_t, a_t, i) \big)^2 \Big], \tag{4}$$

where $Q^{ret}$ is the L-step retrace target (Munos et al., 2016), see the Appendix A.2.2 for details.

**Multitask Policy Improvement 1: Obtaining Non-parametric Policies**  We first find the intermediate policy $q$ by maximizing Equation (3). We obtain a closed-form solution with a non-parametric policy for each task, as

$$q_k(a|s, i) \propto \pi_{\theta_k}(a|s, i) \exp \left( \frac{\hat{Q}(s, a, i)}{\eta} \right), \tag{5}$$

where $\eta$ is a temperature parameter (corresponding to a given bound $\epsilon$) which is optimized alongside the policy optimization (see Appendix A.1.1 for a detailed derivation of the multitask case). As mentioned above, this policy representation is independent of the form of the parametric policy $\pi_{\theta_k}$; i.e. $q$ only depends on $\pi_{\theta_k}$ through its density. This, crucially, makes it easy to employ complicated structured policies (such as the one introduced in Section 3.1). The only requirement here, and in the following steps, is that we must be able to sample from $\pi_{\theta_k}$ and calculate the gradient (w.r.t. $\theta_k$) of its log density (but the sampling process itself need not be differentiable).

**Multitask Policy Improvement 2: Fitting Parametric Policies**  In the second step we fit a policy to the non-parametric distribution obtained from the previous calculation by minimizing the divergence $\mathbb{E}_{s \sim \mathcal{D}, i \sim I}[\text{KL}(q_k(\cdot|s, i) \| \pi_{\theta}(\cdot|s, i))]$. Assuming that we can sample from $q_k$ this step corresponds to maximum likelihood estimation (MLE). Furthermore, we can regularize towards smoothly changing distributions during training – effectively mitigating optimization instabilities and introducing an inductive bias – by limiting the change of the policy (a trust-region constraint). The idea is commonly used in on- as well as in off-policy RL (Schulman et al., 2015; Abdolmaleki et al., 2018b;a). The application to hierarchical policy classes highlights the importance of this constraint as investigated in Section 4.2. Formally, we aim to obtain the solution

$$\theta_{k+1} = \arg \min_{\theta} \mathbb{E}_{s \sim \mathcal{D}, i \sim I} \Big[ \text{KL}\big( q_k(\cdot|s, i) \| \pi_{\theta}(\cdot|s, i) \big) \Big]$$

$$= \arg \max_{\theta} \mathbb{E}_{s \sim \mathcal{D}, i \sim I} \left[ \mathbb{E}_{\pi_{\theta_k}} \left[ \exp(\hat{Q}(s,a,i)/\eta) \log \sum_{o=1}^{M} \pi_{\theta}^{L}(a|s, o) \, \pi_{\theta}^{H}(o|s, i) \right] \right], \tag{6}$$

$$\text{s.t.} \ \mathbb{E}_{s \sim \mathcal{D}, i \sim I} \left[ \text{KL}(\pi_{\theta_k}^{H}(o|s, i) \| \pi_{\theta}^{H}(o|s, i)) + \frac{1}{M} \sum_{o=1}^{M} \text{KL}(\pi_{\theta_k}^{L}(a|s, o) \| \pi_{\theta}^{L}(a|s, o)) \right] < \epsilon_m,$$

where $\epsilon_m$ defines a bound on the change of the new policy. Here we drop constant terms and the negative sign in the second line (turning min into max), and explicitly insert the definition $\pi_{\theta}(a|s, i) = \sum_{o=1}^{M} \pi_L(a|s, o) \pi_H(o|s, i)$, to highlight that we are marginalizing over the high-level choices in this fitting step (since $q$ is not tied to the policy structure). Hence, the update is independent of the specific policy component from which the action was sampled, enabling joint updates of all components. This reduces the variance of the update and also enables efficient off-policy learning. Different approaches can be used to control convergence for both the "high-level" categorical choices and the action choices to change slowly throughout learning. The average KL constraint in Equation (6) is similar in nature to an upper bound on the computationally intractable KL divergence between the two mixture distributions and has been determined experimentally to perform better in practice than simple bounds. In practice, in order to control the change of the high level and low level policies independently we decouple the constraints to be able to set different $\epsilon$ for the means ($\epsilon_{\mu}$), covariances ($\epsilon_{\Sigma}$) and the categorical distribution ($\epsilon_{\alpha}$) in case of a mixture of Gaussian policy. To solve Equation

(6), we first employ Lagrangian relaxation to make it amenable to gradient based optimization and then perform a fixed number of gradient descent steps (using Adam (Kingma and Ba, 2014)); details on this step, as well as an algorithm listing, can be found in the Appendix A.1.2.

# 4 EXPERIMENTS

In the following sections, we investigate the effects of training hierarchical policies in single and multitask domains, finally demonstrating how RHPO can provide compelling benefits for multitask learning in real and simulated robotic manipulation tasks and significantly reduce platform interaction time. In the context of single-task domains from the DeepMind Control Suite (Tassa et al., 2018), we first demonstrate how this type of hierarchy on its own fails to improve performance and that for the model to exploit compositionality, additional incentives for component specialization are required. Subsequently, we introduce suited incentives leading to improved performance and demonstrate that the variety of objectives in multitask domains can serve the same purpose. The evaluation includes experiments on physical hardware with robotic manipulation tasks for the Sawyer arm, emphasizing the importance of data-efficiency.

More details on task hyperparameters as well as the results for additional ablations and all tasks from the multitask domains are provided in the Appendix A.4. Across all tasks, we build on a distributed actor-critic framework (similar to (Espeholt et al., 2018)) with flexible hardware assignment (Buchlovsky et al., 2019) to train all agents, performing critic and policy updates from a replay buffer, which is asynchronously filled by a set of actors. In all figures with error bars, we visualize mean and variance derived from 3 runs.

## 4.1 SIMULATED SINGLE TASK EXPERIMENTS

We consider two high-dimensional tasks for continuous control: humanoid-run and humanoid-stand from Tassa et al. (2018) and compare a flat Gaussian policy to a hierarchical policy, a mixture of Gaussians with three components. We align the update rates of all approaches for fair comparison and to focus the comparison of the algorithm and not its specific implementation [2]. Figure 1 visualizes the results in terms of the number of actor episodes.

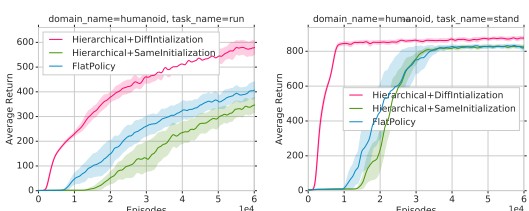

Figure 1: Using a hierarchical policy with different component initialization (red curve) demonstrates benefits over homogeneous initialization as well as the flat Gaussian policy. The plot shows that the simple change in initialization is sufficient to enable component specialization and the correlated improvement in performance.

As can be observed, the hierarchical policy performs comparable to a flat policy with well aligned means and variances for all components as the model fails to decompose the problem. While both the flat and hierarchical policy are initialized with means close to zero, we now include another hierarchical policy with distributed initial means for the three components ranging for all dimensions from minimum to maximum of the allowed action range (here: -1, 0, 1). This simple change suffices to enable component specialization and significantly improved performance.

## 4.2 SIMULATED MULTITASK EXPERIMENTS

We use three simulated multitask scenarios with the Kinova Jaco and Rethink Robotics Sawyer robot arms to test in a variety of conditions. **Pile1**: Here, the seven tasks of interest range from simple reaching for a block over tasks like grasping it, to the final task of stacking the block on top of another block. In addition to the experiments in simulation, which are executed with 5 actors in a distributed setting, the same **Pile1** multitask domain (same rewards and setup) is investigated with a single, physical robot in Section 4.3. We further extend the evaluation towards two more complex multitask domains in simulation. The first extension includes stacking with both blocks on top of the respective

---

[2]In asynchronous RL systems, the update rate of the learner can have a significant impact on the performance if evaluated over actor steps

other block, resulting in a setting with 10 tasks (**Pile2**). And a last domain including harder tasks such as opening a box and placing blocks into this box, consisting of a total of 13 tasks (**Cleanup2**).

We compare RHPO for training **hierarchical** policies against a flat, monolithic policy shared across all tasks which is provided with the additional task id as input (displayed as **Monolithic** in the plot) as well as policies with task dependent heads (displayed as **Independent** in the plots) following (Riedmiller et al., 2018) – both using MPO as the optimizer and a re-implementation of SAC-U using SVG (Heess et al., 2015) (which is related to a version of the option critic (Bacon et al., 2017) without temporal abstraction). The baselines provide the two opposite, naive perspectives on transfer: by using the same monolithic policy across tasks we enable positive as well as negative interference and independent policies prevent policy-based transfer. After experimentally confirming the robustness of RHPO with respect to the number of low-level sub-policies (see Appendix A.8.3), we set $M$ proportional to the number of tasks in each domain.

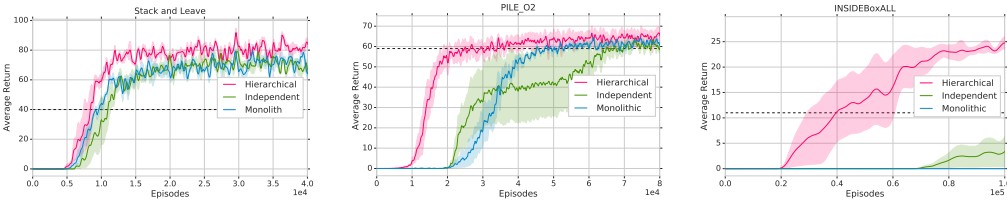

Figure 2: Results for the multitask robotic manipulation experiments in simulation. The dashed line corresponds to the performance of the SVG-based implementation of SAC-U. From left to right: Pile1, Pile2, Cleanup2. We show averages over 3 runs each, with corresponding standard deviation. RHPO outperforms both baselines across all tasks with the benefits increasing for more complex domains.

Figure 2 demonstrates that the hierarchical policy (RHPO) outperforms the monolithic as well as the independent baselines. For simple tasks such as Pile1, the difference is smaller, but the more tasks are trained and the more complex the domain becomes (cf. Pile2 and Cleanup2), the greater is the advantage of composing learned behaviours across tasks. Compared to SVG (Heess et al., 2015), we observe that the baselines based on MPO already result in an improvement, which becomes even bigger with the hierarchical policies. The results across all domains exhibit performance gains for the hierarchical model without the additional incentives from Section 4.1, demonstrating the sufficiency of variety in the training objectives to encourage component specialization and problem decomposition.

### 4.3 PHYSICAL ROBOT EXPERIMENTS

For real-world experiments, data-efficiency is crucial. We perform all experiments in this section relying on a single robot (single actor) – demonstrating the benefits of RHPO in the low data regime. The performed task is the real world version of the Pile1 task described in Section 4.2. The main task objective is to stack one cube onto a second one and move the gripper away from it. We introduce an additional third cube which serves purely as a distractor.

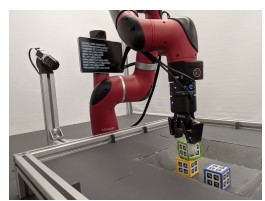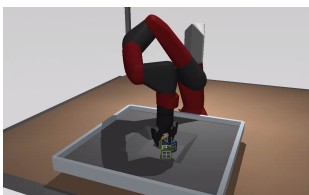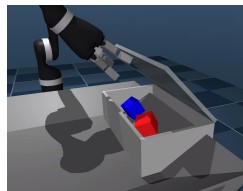

Figure 3: Left: Overview of the real robot setup with the Sawyer robot performing the Pile1 task. Screen pixelated for anonymization. Middle: Simulated Sawyer performing the same task. Right: Cleanup2 setup with the Jaco.

The setup for the experiments consists of a Sawyer robot arm mounted on a table, equipped with a Robotiq 2F-85 parallel gripper. A basket of size $20\text{cm}^2$ in front of the robot contains the three cubes. Three cameras on the basket track the cubes using fiducials (augmented reality tags). As in simulation,

the agent is provided with proprioception information (joint positions, velocities and torques), a wrist sensor's force and torque readings, as well as the cubes' poses – estimated via the fiducials. The agent action is five dimensional and consists of the three Cartesian translational velocities, the angular velocity of the wrist around the vertical axis and the speed of the gripper's fingers.

Figure 4 plots the learning progress on the real robot for two (out of 7) of the tasks, the simple reach tasks and the stack task – which is the main task of interest. Plots for the learning progress of all tasks are given in the appendix A.6. As can be observed, all methods manage to learn the reach task quickly (within about a few thousand episodes) but only RHPO with a hierarchical policy is able to learn the stacking task (taking about 15 thousand episodes to obtain good stacking success), which takes about 8 days of training on the real robot with considerably slower progress for all baselines.

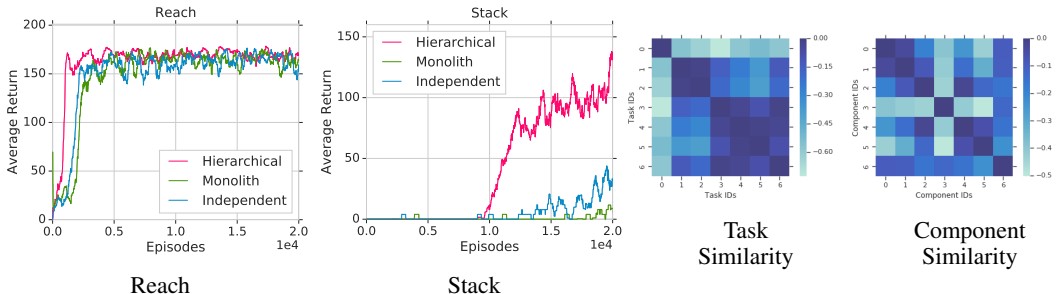

Figure 4: Robot Experiments. Left: While simpler tasks such as reaching are learned with comparable efficiency, the later, more complex tasks are acquired significantly faster with a hierarchical policy. Right: Similarities between tasks (based on their distribution over components) and similarities between components (based on the distribution over tasks which apply them).

In addition, we compute distributions for each component over the tasks which activate it, as well as distributions for each task over which components are being used. For each set of distributions, we determine the Battacharyya distance metric to determine the similarity between tasks and the similarity between components in Figure 4 (right). The plots demonstrate how the components specialise, but also provide a way to investigate our tasks, showing e.g. that the first reach task is fairly independent and that the last four tasks are comparably similar regarding the high-level components applied for their solution.

### 4.3.1 ADDITIONAL ABLATIONS

We perform a series of ablations based on the earlier introduced Pile1 domain, providing additional insights into benefits and shortcomings of RHPO and important factors for robust training. The algorithm is well suited for sequential transfer learning based on solving new tasks with pre-trained low-level components (Appendix A.9). We demonstrate the robustness of RHPO with respect to the number of sub-policies as well as importance of choice of regularization respectively in Appendix A.8.1 and A.8.3. Finally, we ablate over the number of data-generating actors to evaluate all approaches with respect to data rate and illustrate how hierarchical policies are particularly relevant at lower data rates such as given by real-world robotics applications in Appendix A.8.2.

## 5 RELATED WORK

Transfer learning, in particular in the multitask context, has long been part of machine learning (ML) for data-limited domains (Caruana, 1997; Torrey and Shavlik, 2010; Pan and Yang, 2010; Taylor and Stone, 2009). Commonly, it is not straightforward to train a single model jointly across different tasks as the solutions to tasks might not only interfere positively but also negatively (Wang et al., 2018). Preventing this type of forgetting or negative transfer presents a challenge for biological (Singley and Anderson, 1989) as well as artificial systems (French, 1999). In the context of ML, a common scheme is the reduction of representational overlap (French, 1999; Rusu et al., 2016; Wang et al., 2018). Bishop (1994) utilize neural networks to parametrize mixture models for representing multi-modal distributions thus mitigating shortcomings of non-hierarchical approaches. Rosenstein et al. (2005) demonstrate the benefits of hierarchical classification models to limit the impact of negative transfer.

Hierarchical approaches have a long history in the reinforcement learning literature (e.g. Sutton et al., 1999; Dayan and Hinton, 1993). Prior work commonly benefits from combining hierarchy with additional inductive biases such as (Vezhnevets et al., 2017; Nachum et al., 2018a;b; Xie et al., 2018) which employ different rewards for different levels of the hierarchy rather than optimizing a single objective for the entire model as we do. Other works have shown the additional benefits for the stability of training and data-efficiency when sequences of high-level actions are given as guidance during optimization in a hierarchical setting (Shiarlis et al., 2018; Andreas et al., 2017; Tirumala et al., 2019). Instead of introducing additional training signals, we directly investigate the benefits of compositional hierarchy as provided structure for transfer between tasks.

Hierarchical models for probabilistic trajectory modelling have been used for the discovery of behavior abstractions as part of an end-to-end reinforcement learning paradigm (e.g. Teh et al., 2017; Igl et al., 2019; Tirumala et al., 2019; Galashov et al., 2018) where the models act as learned inductive biases that induce the sharing of behavior across tasks. In a vein similar to the presented algorithm, (e.g Heess et al., 2016; Tirumala et al., 2019) share a low-level controller across tasks but modulate the low-level behavior via a continuous embedding rather than picking from a small number of mixture components. In related work Hausman et al. (2018); Haarnoja et al. (2018) learn hierarchical policies with continuous latent variables optimizing the entropy regularized objective.

Similar to our work, the options framework (Sutton et al., 1999; Precup, 2000) supports behavior hierarchies, where the higher level chooses from a discrete set of sub-policies or "options" which commonly are run until a termination criterion is satisfied. The framework focuses on the notion of temporal abstraction. A number of works have proposed practical and scalable algorithms for learning option policies with reinforcement learning (e.g. Bacon et al., 2017; Zhang and Whiteson, 2019; Smith et al., 2018; Riemer et al., 2018; Harb et al., 2018) or criteria for option induction (e.g. Harb et al., 2018; Harutyunyan et al., 2019). Rather than the additional inductive bias of temporal abstraction, we focus on the investigation of composition as type of hierarchy in the context of single and multitask learning while demonstrating the strength of hierarchical composition to lie in domains with strong variation in the objectives - such as in multitask domains. We additionally introduce a hierarchical extension of SVG (Heess et al., 2015), to investigate similarities to work on the option critic (Bacon et al., 2017).

With the use of KL regularization to different ends in RL, work related to RHPO focuses on contextual bandits (Daniel et al., 2016). The algorithm builds on a 2-step EM like procedure to optimize linearly parametrized mixture policies. However, their algorithm has been used only with low dimensional policy representations, and in contextual bandit and other very short horizon settings. Our approach is designed to be applicable to full RL problems in complex domains with long horizons and with high-capacity function approximators such as neural networks. This requires robust estimation of value function approximations, off-policy correction, and additional regularization for stable learning.

## 6 DISCUSSION

We introduce a novel framework to enable robust training and investigation of hierarchical, compositional policies in complex simulated and real-world tasks as well as provide insights into the learning process and its stability. In simulation as well as on real robots, RHPO outperforms baseline methods which either handle tasks independently or utilize implicit sharing. Especially with increasingly complex tasks or limited data rate, as given in real-world applications, we demonstrate hierarchical inductive biases to provide a compelling foundation for transfer learning, reducing the number of environment interactions significantly and often leading to more robust learning as well as improved final performance. For single tasks with a single training objective all components can remain aligned, preventing problem decomposition and the hierarchical policy replicates a flat policy. Performance improvements appear only when the individual components specialize, either via variety in the training objectives or additional incentives. Furthermore, as demonstrated in Appendix A.9, a pre-trained set of specialized components can notably improve performance when learning new tasks. One important next step is identifying how to optimize a basis set of components which transfers well to a wide range of tasks

Since with mixture distributions, we are able to marginalize over components when optimizing the weighted likelihood over action samples in Equation 6, the extension towards multiple levels of hierarchy is trivial but can provide a valuable direction for practical future work. While this approach partially mitigates negative interference between tasks in a parallel multitask learning scenario, addressing catastrophic inference in sequential settings remains a challenge.

We believe that especially in domains with consistent agent embodiment and high costs for data generation learning tasks jointly and information sharing is imperative. RHPO combines several ideas that we believe will be important: multitask learning with hierarchical and compositional policy representations, robust optimization, and efficient off-policy learning. Although we have found this particular combination of components to be very effective we believe it is just one instance of – and step towards – a spectrum of efficient learning architectures that will unlock further applications of RL both in simulation and, importantly, on physical hardware.

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

## A  APPENDIX

### A.1  ADDITIONAL DERIVATIONS

In this section we explain the detailed derivations for training hierarchical policies parameterized as a mixture of Gaussians.

#### A.1.1  OBTAINING NON-PARAMETRIC POLICIES

In each policy improvement step, to obtain non-parametric policies for a given state and task distribution, we solve the following program:

$$\max_q \mathbb{E}_{\mu(s),i\sim I}\big[\mathbb{E}_{q(a|s,i)}\big[\hat{Q}(s,a,i)\big]\big]$$
$$s.t.\mathbb{E}_{\mu(s),i\sim I}\big[\mathrm{KL}(q(\cdot|s,i),\pi(\cdot|s,i,\theta_t))\big] < \epsilon$$
$$s.t.\mathbb{E}_{\mu(s),i\sim I}\big[\mathbb{E}_{q(a|s)}\big[1\big]\big] = 1.$$

To make the following derivations easier to follow we open up the expectations, writing them as integrals explicitly. For this purpose let us define the joint distribution over states $s \sim \mu(s)$ together with randomly sampled tasks $i \sim I$ as $\mu(s,i) = p(s|\mathcal{D})\mathcal{U}(i \in I)$, where $\mathcal{U}$ denotes the uniform distribution over possible tasks. This allows us to re-write the expectations that include the corresponding distributions, i.e. $\mathbb{E}_{\mu(s),i\sim I}[1] = \mathbb{E}_{\mu(s,i)}[1]$, but again, note that $i$ here is not necessarily the task under which $s$ was observed. We can then write the Lagrangian equation corresponding to the above described program as

$$L(q,\eta,\gamma) = \int \mu(s,i) \int q(a|s,i)\hat{Q}(s,a,i)\,\mathrm{d}a\,\mathrm{d}s\,\mathrm{d}i$$
$$+ \eta\left(\epsilon - \int \mu(s,i)\int q(a|s,i)\log\frac{q(a|s,i)}{\pi(a|s,i,\theta_t)\,\mathrm{d}a\,\mathrm{d}s\,\mathrm{d}i}\right)$$
$$+ \gamma\left(1 - \iint \mu(s,i)q(a|s,i)\,\mathrm{d}a\,\mathrm{d}s\,\mathrm{d}i\right).$$

Next we maximize the Lagrangian $L$ w.r.t the primal variable $q$. The derivative w.r.t $q$ reads,

$$\partial qL(q,\eta,\gamma) = \hat{Q}(a,s,i) - \eta\log q(a|s,i)$$
$$+ \eta\log\pi(a|s,i,\theta_t) - \eta - \gamma.$$

Setting it to zero and rearranging terms we obtain

$$q(a|s,i) = \pi(a|s,i,\theta_t)\exp\left(\frac{\hat{Q}(s,a,i)}{\eta}\right)\exp\left(-\frac{\eta+\gamma}{\eta}\right).$$

However, the last exponential term is a normalization constant for $q$. Therefore we can write,

$$\exp\left(\frac{\eta+\gamma}{\eta}\right) = \int \pi(a|s,i,\theta_t)\exp\left(\frac{Q(s,a,i)}{\eta}\right)\mathrm{d}a$$

$$\frac{\eta+\gamma}{\eta} = \log\left(\int \pi(a|s,i,\theta_t)\exp\left(\frac{Q(s,a,i)}{\eta}\right)\mathrm{d}a\right). \tag{7}$$

Now, to obtain the dual function $g(\eta)$, we plug in the solution to the KL constraint term (second term) of the Lagrangian which yields

$$
\begin{aligned}
L(q, \eta, \gamma) = &\int \mu(s, i) \int q(a|s, i) Q(s, a, i) \, \mathrm{d}a \, \mathrm{d}s \\
&+ \eta \left( \epsilon - \int \mu(s, i) \int q(a|s, i) \log \frac{\pi(a|s, i, \theta_t) \exp\left(\frac{Q(s, a, i)}{\eta}\right) \exp\left(-\frac{\eta + \gamma}{\eta}\right)}{\pi(a|s, i, \theta_t)} \, \mathrm{d}a \, \mathrm{d}s \, \mathrm{d}i \right) \\
&+ \gamma \left( 1 - \iint \mu(s, i) q(a|s, i) \, \mathrm{d}a \, \mathrm{d}s \, \mathrm{d}i \right).
\end{aligned}
$$

After expanding and rearranging terms we get

$$
\begin{aligned}
L(q, \eta, \gamma) = &\int \mu(s, i) \int q(a|s, i) Q(s, a, i) \, \mathrm{d}a \, \mathrm{d}s \, \mathrm{d}i \\
&- \eta \int \mu(s, i) \int q(a|s, i) \left[ \frac{Q(s, a, i)}{\eta} + \log \pi(a|s, i; \theta_t) - \frac{\eta + \gamma}{\eta} \right] \mathrm{d}a \, \mathrm{d}s \, \mathrm{d}i \\
&+ \eta \epsilon + \eta \int \mu(s, i) \int q(a|s, i) \log \pi(a|s, i; \theta_t) \, \mathrm{d}a \, \mathrm{d}s \, \mathrm{d}i \\
&+ \gamma \left( 1 - \iint \mu(s, i) q(a|s, i) \, \mathrm{d}a \, \mathrm{d}s \, \mathrm{d}i \right).
\end{aligned}
$$

Most of the terms cancel out and after rearranging the terms we obtain

$$
L(q, \eta, \gamma) = \eta \epsilon + \eta \int \mu(s, i) \frac{\eta + \gamma}{\eta} \, \mathrm{d}s \, \mathrm{d}i.
$$

Note that we have already calculated the term inside the integral in Equation 7. By plugging in equation 7 we obtain the dual function

$$
g(\eta) = \eta \epsilon + \eta \int \mu(s, i) \log \left( \int \pi(a|s, i, \theta_t) \exp\left( \frac{Q(s, a, i)}{\eta} \right) \mathrm{d}a \right) \mathrm{d}s \, \mathrm{d}i, \tag{8}
$$

which we can minimize with respect to $\eta$ based on samples from the replay buffer.

### A.1.2 EXTENDED UPDATE RULES FOR FITTING A MIXTURE OF GAUSSIANS

After obtaining the non parametric policies, we fit a parametric policy to samples from said non-parametric policies – effectively employing using maximum likelihood estimation with additional regularization based on a distance function $\mathcal{T}$, i.e,

$$
\begin{aligned}
\theta^{(k+1)} &= \arg \min_\theta \mathbb{E}_{s \sim \mathcal{D}, i \sim I} \left[ \mathrm{KL}\big( q_k(\cdot|s, i) \| \pi_\theta(\cdot|s, i) \big) \right] \\
&= \arg \min_\theta \mathbb{E}_{s \sim \mathcal{D}, i \sim I, a \sim q(\cdot|s, i)} \left[ - \log \pi_\theta(a|s, i) \right], \\
&= \arg \max_\theta \mathbb{E}_{s \sim \mathcal{D}, i \sim I, a \sim q(\cdot|s, i)} \left[ \log \pi_\theta(a|s, i) \right], \\
&\text{s.t. } \mathbb{E}_{s \sim \mathcal{D}, i \sim I} \left[ \mathcal{T}(\pi_{\theta_k}(\cdot|s, i) | \pi_\theta(\cdot|s, i)) \right] < \epsilon,
\end{aligned} \tag{9}
$$

where $\mathcal{D}$ is an arbitrary distance function to evaluate the change of the new policy with respect to a reference/old policy, and $\epsilon$ denotes the allowed change for the policy. To make the above objective amenable to gradient based optimization we employ Lagrangian Relaxation, yielding the following primal:

$$
\max_\theta \min_{\alpha > 0} L(\theta, \alpha) = \mathbb{E}_{s \sim \mathcal{D}, i \sim I, a \sim q(\cdot|s, i)} \left[ \log \pi_\theta(a|s, i) \right] + \alpha \Big( \epsilon - \mathbb{E}_{s \sim \mathcal{D}, , i \sim I} \left[ \mathcal{T}(\pi_{\theta_k}(\cdot|s, i), \pi_\theta(\cdot|s, i)) \right] \Big). \tag{10}
$$

We solve for $\theta$ by iterating the inner and outer optimization programs independently: We fix the parameters $\theta$ to their current value and optimize for the Lagrangian multipliers (inner minimization) and then we fix the Lagrangian multipliers to their current value and optimize for $\theta$ (outer maximization). In practice we found that it is effective to simply perform one gradient step each in inner and outer optimization for each sampled batch of data.

The optimization given above is general, i.e. it works for any general type of policy. As described in the main paper, we consider hierarchical policies of the form

$$\pi_\theta(a|s,i) = \sum_{o=1}^{M} \pi_\theta^L(a|s,o)\,\pi_\theta^H(o|s,i). \tag{11}$$

In particular, in all experiments we made use of a mixture of Gaussians parametrization, where the high level policy $\pi_\theta^H$ is a categorical distribution over low level $\pi_\theta^L$ Gaussian policies, i.e,

$$\pi_\theta(a|s,i) = \pi(a|\{\alpha_\theta^j, \mu_\theta^j, \Sigma_\theta^j\}(s)_{j=1\ldots M}) = \sum_{j=1}^{M} \alpha_\theta^j(s,i)\mathcal{N}(\mu_\theta^j(s), \Sigma_\theta^j(s))$$

$$\forall s \sum_{j=1}^{M} \alpha^j(s,i) = 1, \text{ and, } \alpha^j(s,i) > 0$$

where $j$ denote the index of components and $\alpha$ is the high level policy $\pi^H$ assigning probabilities to each mixture component for a state s given the task and the low level policies are all Gaussian. Here $\alpha^j$s are the probabilities for a categorical distribution over the components.

We also define the following distance function between old and new mixture of Gaussian policies

$$\mathcal{T}(\pi_{\theta_k}(\cdot|s,i), \pi_\theta(\cdot|s,i)) = \mathcal{T}_H(s,i) + \mathcal{T}_L(s)$$

$$\mathcal{T}_H(s,i) = \text{KL}(\text{Cat}(\{\alpha_{\theta_k}^j(s,i)\}_{j=1\ldots M})\|\text{Cat}(\{\alpha_\theta^j(s,i)\}_{j=1\ldots M}))$$

$$\mathcal{T}_L(s) = \frac{1}{M}\sum_{j=1}^{M} \text{KL}(\mathcal{N}(\mu_{\theta_k}^j(s), \Sigma_{\theta_k}^j(s))\|\mathcal{N}(\mu_\theta^j(s), \Sigma_\theta^j(s)))$$

where $\mathcal{T}_H$ evaluates the KL between categorical distributions and $\mathcal{T}_L$ corresponds to the average KL across Gaussian components, as also described in the main paper (c.f. Equation 5 in the main paper).

In order to bound the change of categorical distributions, means and covariances of the components independently – which makes it easy to control the convergence of the policy and which can prevent premature convergence as argued in Abdolmaleki et al. (2018a) – we separate out the following three intermediate policies

$$\pi_\theta^\Sigma(a|s,i) = \pi(a|\{\alpha_{\theta_k}^j, \mu_{\theta_k}^j, \Sigma_\theta^j\}(s,i)_{j=1\ldots M})$$
$$\pi_\theta^\mu(a|s,i) = \pi(a|\{\alpha_{\theta_k}^j, \mu_\theta^j, \Sigma_{\theta_k}^j\}(s,i)_{j=1\ldots M})$$
$$\pi_\theta^\alpha(a|s,i) = \pi(a|\{\alpha_\theta^j, \mu_{\theta_k}^j, \Sigma_{\theta_k}^j\}(s,i)_{j=1\ldots M})$$

Which yields the following final optimization program

$$\theta^{(k+1)} = \arg\max_\theta \mathbb{E}_{s\sim\mathcal{D}, i\sim I, a\sim q(\cdot|s,i)}\Big[\log\pi_\theta^\mu(a|s,i) + \log\pi_\theta^\Sigma(a|s,i) + \log\pi_\theta^\alpha(a|s,i)\Big],$$

$$\text{s.t. } \mathbb{E}_{s\sim\mathcal{D}, i\sim I}\Big[\mathcal{T}(\pi_{\theta_k}(a|s,i)|\pi_\theta^\mu(a|s,i))\Big] < \epsilon_\mu,$$

$$\text{s.t. } \mathbb{E}_{s\sim\mathcal{D}, i\sim I}\Big[\mathcal{T}(\pi_{\theta_k}(a|s,i)|\pi_\theta^\Sigma(a|s,i))\Big] < \epsilon_\Sigma,$$

$$\text{s.t. } \mathbb{E}_{s\sim\mathcal{D}, i\sim I}\Big[\mathcal{T}(\pi_{\theta_k}(a|s,i)|\pi_\theta^\alpha(a|s,i))\Big] < \epsilon_\alpha,$$

$$\tag{12}$$

This decoupling allows us to set different $\epsilon$ values for the change of means, covariances and categorical probabilities, i.e., $\epsilon_\mu, \epsilon_\Sigma, \epsilon_\alpha$. Different $\epsilon$ lead to different learning rates. We always set a much smaller epsilon for the covariance and categorical than for the mean. The intuition is that while we would like the distribution to converge quickly in the action space, we also want to keep the exploration both locally (via the covariance matrix) and globally (via the high level categorical distribution) to avoid premature convergence.

## A.2 Algorithmic Details

### A.2.1 Pseudocode for the full procedure

We provide a pseudo-code listing of the full optimization procedure – and the asynchronous data gathering – performed by RHPO in Algorithm 1 and 2. The implementation relies on Sonnet (Reynolds et al., 2017) and TensorFlow (Abadi et al., 2015).

---

**Algorithm 1** Asynchronous Learner

---

**Input:** $N_{steps}$ number of update steps, $N_{\text{targetUpdate}}$ update steps between target update, $N_s$ number of action samples per state, KL regularization parameters $\epsilon$, initial parameters for $\pi$, $\eta$ and $\phi$
initialize N = 0
**while** $k \leq N_{\text{steps}}$ **do**
    **for** $k$ in $[0...N_{\text{targetUpdate}}]$ **do**
        sample a batch of trajectories $\tau$ from replay buffer $B$
        sample $N_s$ actions from $\pi_{\theta_k}$ to estimate expectations below
        // compute mean gradients over batch for policy, Lagrangian multipliers and Q-function
        $\delta_\pi \leftarrow -\nabla_\theta \sum_{s_t \in \tau} \sum_{j=1}^{N_s} \exp\left(\frac{Q(s_t,a_j,i)}{\eta}\right) \log \pi_\theta(a_j|s_t,i)$ following Eq. 6
        $\delta_\eta \leftarrow \nabla_\eta g(\eta) = \nabla_\eta \eta \epsilon + \eta \sum_{s_t \in \tau} \log \frac{1}{N_s} \sum_{j=1}^{N_s} \exp\left(\frac{Q(s_t,a_j,i)}{\eta}\right)$ following Eq. 8
        $\delta_Q \leftarrow \nabla_\phi \sum_{i \sim I} \sum_{(s_t,a_t) \in \tau} \left(\hat{Q}_\phi(s_t,a_t,i) - Q^{\text{ret}}\right)^2$ with $Q^{\text{ret}}$ following Eq. 4
        // apply gradient updates
        $\pi_{\theta_{k+1}} = $ optimizer_update$(\pi, \delta_\pi)$,
        $\eta = $ optimizer_update$(\eta, \delta_\eta)$
        $\hat{Q}_\phi = $ optimizer_update$(\hat{Q}_\phi, \delta_Q)$
        $k = k + 1$
    **end for**
    // update target networks
    $\pi' = \pi, Q' = Q$
**end while**

---

### A.2.2 Details on the policy improvement step

As described in the main paper we consider the same setting as scheduled auxiliary control setting (SAC-X) (Riedmiller et al., 2018) to perform policy improvement (with uniform random switches between tasks every N steps within an episode, the SAC-U setting).

Given a replay buffer containing data gathered from all tasks, where for each trajectory snippet $\tau = \{(s_0, a_0, R_0), \ldots, (s_L, a_L, R_L)\}$ we record the rewards for all tasks $R_t = [r_{i_1}(s_t, a_t), \ldots, r_{i_{|I|}}(s_t, a_t)]$ as a vector in the buffer, we define the retrace objective for learning $\hat{Q}$, parameterized via $\phi$, following Riedmiller et al. (2018) as

$$\min_\phi L(\phi) = \sum_{i \sim I} \mathbb{E}_{\tau \sim \mathcal{D}} \left[ \left( r_i(s_t, a_t) + \gamma Q^{ret}(s_{t+1}, a_{t+1}, i) - \hat{Q}_\phi(s_t, a_t, i) \right)^2 \right], \text{ with}$$

$$Q^{\text{ret}}(s_t, a_t, i) = \sum_{j=t}^{\infty} \left( \gamma^{j-t} \prod_{k=t}^{j} c_k \right) \left[ \delta_Q(s_j, s_{j+1}) \right], \tag{13}$$

$$\delta_Q(s_j, s_{j+1}) = r_i(s_j, a_j) + \gamma \mathbb{E}_{\pi_{\theta_k}(a|s_{j+1},i)}, [\hat{Q}_{\phi'}(s_{j+1}, \cdot, i; \phi')] - \hat{Q}_{\phi'}(s_j, a_j, i),$$

where the importance weights are defined as $c_k = \min\left(1, \frac{\pi_{\theta_k}(a_k|s_k,i)}{b(a_k|s_k)}\right)$, with $b(a_k|s_k)$ denoting an arbitrary behavior policy; in particular this will be the policy for the executed tasks during an

---

**Algorithm 2** Asynchronous Actor

---

**Input:** $N_{\text{trajectories}}$ number of total trajectories requested, $T$ steps per episode, $\xi$ scheduling period
initialize N = 0
**while** $N < N_{\text{trajectories}}$ **do**
    fetch parameters $\theta$
    // collect new trajectory from environment
    $\tau = \{\}$
    **for** $t$ in $[0...T]$ **do**
        **if** $t \pmod \xi \equiv 0$ **then**
            // sample active task from uniform distribution
            $i_{act} \sim I$
        **end if**
        $a_t \sim \pi_\theta(\cdot | s_t, i_{act})$
        // execute action and determine rewards for all tasks
        $\bar{r} = [r_{i_1}(s_t, a_t), \ldots, r_{i_{|I|}}(s_t, a_t)]$
        $\tau \leftarrow \tau \cup \{(s_t, a_t, \bar{r}, \pi_\theta(a_0 | s_t, i_{act}))\}$
    **end for**
    send batch trajectories $\tau$ to replay buffer
    $N = N + 1$
**end while**

---

episode as in (Riedmiller et al., 2018). Note that, in practice, we truncate the infinite sum after $L$ steps, bootstrapping with $\hat{Q}$. We further perform optimization of Equation (4) via gradient descent and make use of a target network (Mnih et al., 2015), denoted with parameters $\phi'$, which we copy from $\phi$ after a couple of gradient steps. We reiterate that, as the state-action value function $\hat{Q}$ remains independent of the policy's structure, we are able to utilize any other off-the-shelf Q-learning algorithm such as TD(0) (Sutton, 1988).

Given that we utilize the same policy evaluation mechanism as SAC-U it is worth pausing here to identify the differences between SAC-U and our approach. The main difference is in the policy parameterization: SAC-U used a monolithic policy for each task $\pi(a|s, i)$ (although a neural network with shared components, potentially leading to some implicit task transfer, was used). Furthermore, we perform policy optimization based on MPO instead of using stochastic value gradients (SVG (Heess et al., 2016)). We can thus recover a variant of plain SAC-U using MPO if we drop the hierarchical policy parameterization, which we employ in the single task experiments in the main paper.

### A.2.3 NETWORK ARCHITECTURES

To represent the Q-function in the multitask case we use the network architecture from SAC-X (see right sub-figure in Figure 5). The proprioception of the robot, the features of the objects and the actions are fed together in a torso network. At the input we use a fully connected first layer of 200 units, followed by a layer normalization operator, an optional tanh activation and another fully connected layer of 200 units with an ELU activation function. The output of this torso network is shared by independent head networks for each of the tasks (or intentions, as they are called in the SAC-X paper). Each head has two fully connected layers and outputs a Q-value for this task, given the input of the network. Using the task identifier we then can compute the Q value for a given sample by discrete selection of the according head output.

While we use the network architecture for the Q function for all multitask experiments, we investigate different architectures for the policy in this paper. The original SAC-X policy architecture is shown in Figure 5 (left sub-figure). The main structure follows the same basic principle that we use in the Q function architecture. The only difference is that the heads compute the required parameters for the policy distribution we want to use (see subsection A.2.4). This architecture is referenced as the independent heads (or task dependent heads).

The alternatives we investigate in this paper are the monolithic policy architecture (see Figure 6, left sub-figure) and the hierarchical policy architecture (see Figure 6, right sub-figure). For the monolithic policy architecture we reduce the original policy architecture basically to one head and

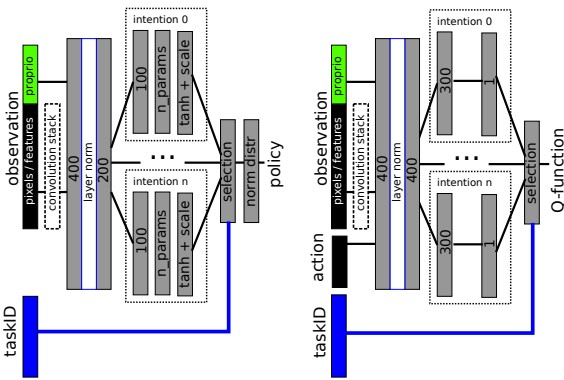

Figure 5: Schematics of the fully connected networks in SAC-X. While we use the Q-function (right sub-figure) architecture in all multitask experiments, we investigate variations of the policy architecture (left sub-figure) in this paper (see Figure 6).

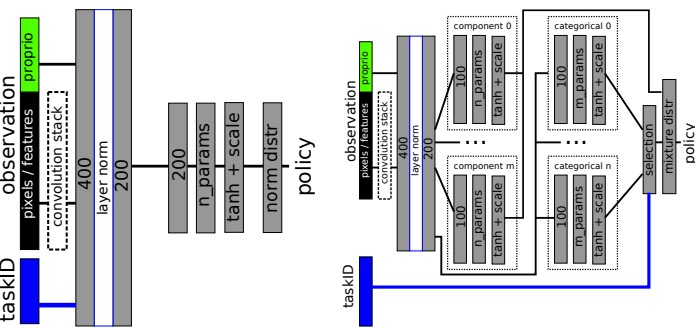

Figure 6: Schematics of the alternative multitask policy architectures used in this paper. Left sub-figure: the monolithic architecture; Right sub-figure: the hierarchical architecture.

append the task-id as a one-hot encoded vector to the input. For the hierarchical architecture, we build on the same torso and create a set of networks parameterizing the Gaussians which are shared across tasks and a task-specific network to parameterize the categorical distribution for each task. The final mixture distribution is task-dependent for the high-level controller but task-independent for the low-level policies.

### A.2.4 ALGORITHM HYPERPARAMETERS

In this section we outline the details on the hyperparameters used for RHPO and baselines in both single task and multitask experiments. All experiments use feed-forward neural networks. We consider a flat policy represented by a Gaussian distribution and a hierarchical policy represented by a mixture of Gaussians distribution. The flat policy is given by a Gaussian distribution with a diagonal covariance matrix, i.e, $\pi(a|s, \theta) = \mathcal{N}(\mu, \Sigma)$. The neural network outputs the mean $\mu = \mu(s)$ and diagonal Cholesky factors $A = A(s)$, such that $\Sigma = AA^T$. The diagonal factor $A$ has positive diagonal elements enforced by the softplus transform $A_{ii} \leftarrow \log(1 + \exp(A_{ii}))$ to ensure positive definiteness of the diagonal covariance matrix. Mixture of Gaussian policy has a number of Gaussian components as well as a categorical distribution for selecting the components. The neural network outputs the Gaussian components based on the same setup described above for a single Gaussian and outputs the logits for representing the categorical distribution. Tables 1 show the hyperparameters we used for the single tasks experiments. We found layer normalization and a hyperbolic tangent ($tanh$) on the layer following the layer normalization are important for stability of the algorithms. For RHPO the most important hyperparameters are the constraints in Step 1 and Step 2 of the algorithm.

| Hyperparameters | Hierarchical | Single Gaussian |
|---|---|---|
| Policy net | 200-200-200 | 200-200-200 |
| Number of actions sampled per state | 10 | 10 |
| Q function net | 500-500-500 | 500-500-500 |
| Number of components | 3 | NA |
| $\epsilon$ | 0.1 | 0.1 |
| $\epsilon_\mu$ | 0.0005 | 0.0005 |
| $\epsilon_\Sigma$ | 0.00001 | 0.00001 |
| $\epsilon_\alpha$ | 0.0001 | NA |
| Discount factor ($\gamma$) | 0.99 | 0.99 |
| Adam learning rate | 0.0002 | 0.0002 |
| Replay buffer size | 2000000 | 2000000 |
| Target network update period | 250 | 250 |
| Batch size | 256 | 256 |
| Activation function | elu | elu |
| Layer norm on first layer | Yes | Yes |
| Tanh on output of layer norm | Yes | Yes |
| Tanh on input actions to Q-function | Yes | Yes |
| Retrace sequence length | 10 | 10 |

Table 1: Hyperparameters - Single Task

| Hyperparameters | Hierarchical | Independent | Monolith |
|---|---|---|---|
| Policy torso (shared across tasks) | 400-200 | | |
| Policy task-dependent heads | 100 (high-level controller) | 100 | NA |
| Policy shared heads | 100 (per mixture component) | NA | 200 |
| Number of action samples | 20 | | |
| Q function torso (shared across tasks) | 400-400 | | |
| Q function head (per task) | 300 | | |
| Number of components | number of tasks | NA | NA |
| Discount factor ($\gamma$) | 0.99 | | |
| Replay buffer size | 1e6 * number of tasks | | |
| Target network update period | 500 | | |
| Batch size | 256 (512 for Pile1) | | |

Table 2: Hyperparameters - Multitask. Values are taken from the single task experiments with the above mentioned changes.

### A.3 ADDITIONAL DETAILS ON THE SAC-U WITH SVG BASELINE

For the SAC-U baseline we used a re-implementation of the method from (Riedmiller et al., 2018) using SVG (Heess et al., 2015) for optimizing the policy. Concretely we use the same basic network structure as for the "Monolithic" baseline with MPO and parameterize the policy as

$$\pi_\theta = \mathcal{N}(\mu_\theta(s, i), \sigma_\theta^2(s, i)I),$$

where $I$ denotes the identity matrix and $\sigma_\theta(s, i)$ is computed from the network output via a softplus activation function.

Together with entropy regularization, as described in (Riedmiller et al., 2018) the policy can be optimized via gradient ascent, following the reparameterized gradient for a given states sampled from the replay:

$$\nabla_\theta \mathbb{E}_{\pi_\theta(a|s,i)}[\hat{Q}(a, s, i)] + \alpha \mathrm{H}\Big(\pi_\theta(a|s, i)\Big), \tag{14}$$

which can be computed, using the reparameterization trick, as

$$\mathbb{E}_{\zeta \sim \mathcal{N}(0,I)}[\nabla_\theta g_\theta(s, i, \zeta) \nabla_g Q(g_\theta(s, i, \zeta), s, i)] + \alpha \nabla_\theta \mathrm{H}\Big(\pi_\theta(a|s, i)\Big), \tag{15}$$

where $g_\theta(s, i, \zeta) = \mu_\theta(s, i) + \sigma_\theta(\mathbf{s}) * \zeta$ is now a deterministic function of a sample from the standard multivariate normal distribution. See e.g. Heess et al. (2015) (for SVG) as well as Kingma and Welling (2014) (for the reparameterization trick) for a detailed explanation.

## A.4 DETAILS ON THE EXPERIMENTAL SETUP

### A.4.1 SIMULATION (SINGLE- AND MULTITASK)

For the simulation of the robot arm experiments the numerical simulator MuJoCo [3] was used – using a model we identified from the real robot setup.

We run experiments of length 2 - 7 days for the simulation experiments (depending on the task) with access to 2-5 recent CPUs with 32 cores each (depending on the number of actors) and 2 recent NVIDIA GPUs for the learner. Computation for data buffering is negligible.

### A.4.2 REAL ROBOT MULTITASK

Compared to simulation where the ground truth position of all objects is known, in the real robot setting, three cameras on the basket track the cube using fiducials (augmented reality tags).

For safety reasons, external forces are measured at the wrist and the episode is terminated if a threshold of 20N on any of the three principle axes is exceeded (this is handled as a terminal state with reward 0 for the agent), adding further to the difficulty of the task.

The real robot setup differs from the simulation in the reset behaviour between episodes, since objects need to be physically moved around when randomizing, which takes a considerable amount of time. To keep overhead small, object positions are randomized only every 25 episodes, using a hand-coded controller. Objects are also placed back in the basket if they were thrown out during the previous episode. Other than that, objects start in the same place as they were left in the previous episode. The robot's starting pose is randomized each episode, as in simulation.

## A.5 TASK DESCRIPTIONS

### A.5.1 PILE1

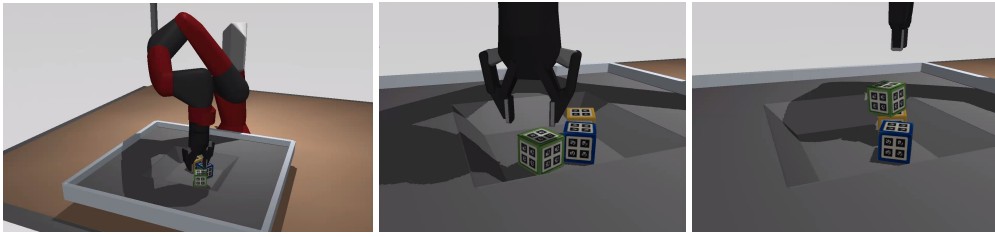

Figure 7: Sawyer Set-Up.

For this task we have a real setup and a MuJoCo simulation that are well aligned. It consists of a Sawyer robot arm mounted on a table and equipped with a Robotiq 2F-85 parallel gripper. In front of the robot there is a basket of size 20x20 cm which contains three cubes with an edge length of 5 cm (see Figure 7).

The agent is provided with proprioception information for the arm (joint positions, velocities and torques), and the tool center point position computed via forward kinematics. For the gripper, it receives the motor position and velocity, as well as a binary grasp flag. It also receives a wrist sensor's force and torque readings. Finally, it is provided with the cubes' poses as estimated via the fiducials, and the relative distances between the arm's tool center point and each object. At each time step, a history of two previous observations is provided to the agent, along with the last two joint control commands, in order to account for potential communication delays on the real robot. The observation space is detailed in Table 4.

---

[3]MuJoCo: see www.mujoco.org

Table 3: Action space for the Sawyer experiments.

| Entry | Dimensions | Unit | Range |
|---|---|---|---|
| Translational Velocity in x, y, z | 3 | m/s | [-0.07, 0.07] |
| Wrist Rotation Velocity | 1 | rad/s | [-1, 1] |
| Finger speed | 1 | tics/s | [-255, 255] |

Table 4: Observation used in the experiments with the Sawyer arm. An object's pose is represented as its world coordinate position and quaternion. In the table, $m$ denotes meters, $rad$ denotes radians, and $q$ refers to a quaternion in arbitrary units ($au$).

| Entry | Dimensions | Unit |
|---|---|---|
| Joint Position (Arm) | 7 | rad |
| Joint Velocity (Arm) | 7 | rad/s |
| Joint Torque (Arm) | 7 | Nm |
| Joint Position (Hand) | 1 | rad |
| Joint Velocity (Hand) | 1 | tics/s |
| Force-Torque (Wrist) | 6 | N, Nm |
| Binary Grasp Sensor | 1 | au |
| TCP Pose | 7 | m, au |
| Last Control Command (Joint Velocity) | 8 | rad/s |
| Green Cube Pose | 7 | m, au |
| Green Cube Relative Pose | 7 | m, au |
| Yellow Cube Pose | 7 | m, au |
| Yellow Cube Relative Pose | 7 | m, au |
| Blue Cube Pose | 7 | m, au |
| Blue Cube Relative Pose | 7 | m, au |

The robot arm is controlled in Cartesian mode at 20Hz. The action space for the agent is 5-dimensional, as detailed in Table 3. The gripper movement is also restricted to a cubic volume above the basket using virtual walls.

For the Pile1 experiment we use 7 different task to learn, following the SAC-X principles. The first 6 tasks are seen as auxiliary tasks that help to learn the final task (STACK_AND_LEAVE(G, Y)) of stacking the green cube on top of the yellow cube. Overview of the used tasks:

- *REACH(G)*: $stol(d(TCP, G), 0.02, 0.15)$:
  Minimize the distance of the TCP to the green cube.

- *GRASP*:
  Activate grasp sensor of gripper ("inward grasp signal" of Robotiq gripper)

- *LIFT(G)*: $slin(G, 0.03, 0.10)$
  Increase z coordinate of an object more than 3cm relative to the table.

- *PLACE_WIDE(G, Y)*: $stol(d(G, Y + [0, 0, 0.05]), 0.01, 0.20)$
  Bring green cube to a position 5cm above the yellow cube.

- *PLACE_NARROW(G, Y)*: $stol(d(G, Y + [0, 0, 0.05]), 0.00, 0.01)$:
  Like PLACE_WIDE(G, Y) but more precise.

- *STACK(G, Y)*: $btol(d_{xy}(G, Y), 0.03) * btol(d_z(G, Y) + 0.05, 0.01) * (1 - GRASP)$
  Sparse binary reward for bringing the green cube on top of the yellow one (with 3cm tolerance horizontally and 1cm vertically) and disengaging the grasp sensor.

- *STACK_AND_LEAVE(G, Y)*: $stol(d_z(TCP, G) + 0.10, 0.03, 0.10) * STACK(G, Y)$
  Like STACK(G, Y), but needs to move the arm 10cm above the green cube.

Table 5: Action space used in the experiments with the Kinova Jaco Arm.

| Entry | Dimensions | Unit | Range |
|---|---|---|---|
| Joint Velocity (Arm) | 6 | rad/sec | [-0.8, 0.8] |
| Joint Velocity (Hand) | 3 | rad/sec | [-0.8, 0.8] |

Let $d(o_i, o_j)$ be the distance between the reference of two objects (the reference of the cubes are the center of mass, TCP is the reference of the gripper), and let $d_A$ be the distance only in the dimensions denoted by the set of axes $A$. We can define the reward function details by:

$$stol(v, \epsilon, r) = \begin{cases} 1 & \text{iff } |v| < \epsilon \\ 1 - tanh^2(\frac{atanh(\sqrt{0.95})}{r}|v|) & \text{else} \end{cases} \tag{16}$$

$$slin(v, \epsilon_{min}, \epsilon_{max}) = \begin{cases} 0 & \text{iff } v < \epsilon_{min} \\ 1 & \text{iff } v > \epsilon_{max} \\ \frac{v - \epsilon_{min}}{\epsilon_{max} - \epsilon_{min}} & \text{else} \end{cases} \tag{17}$$

$$btol(v, \epsilon) = \begin{cases} 1 & \text{iff } |v| < \epsilon \\ 0 & \text{else} \end{cases} \tag{18}$$

### A.5.2 PILE2

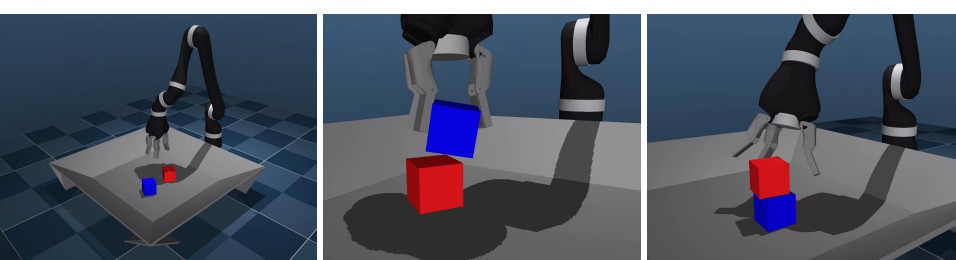

Figure 8: The *Pile2* set-up in simulation with two main tasks: The first is to stack the blue on the red cube, the second is to stack the red on the blue cube.

For the *Pile2* task, taken from Riedmiller et al. (2018), we use a different robot arm, control mode and task setup to emphasize that RHPO's improvements are not restricted to cartesian control or a specific robot and that the approach also works for multiple external tasks.

Here, the agent controls a simulated Kinova Jaco robot arm, equipped with a Kinova KG-3 gripper. The robot faces a 40 x 40 cm basket that contains a red cube and a blue cube. Both cubes have an edge length of 5 cm (see Figure 8). The agent is provided with proprioceptive information for the arm and the fingers (joint positions and velocities) as well as the tool center point position (TCP) computed via forward kinematics. Further, the simulated gripper is equipped with a touch sensor for each of the three fingers, whose value is provided to the agent as well. Finally, the agent receives the cubes' poses, their translational and rotational velocities and the relative distances between the arm's tool center point and each object. Neither observation nor action history is used in the *Pile2* experiments. The cubes are spawned at random on the table surface and the robot hand is initialized randomly above the table-top with a height offset of up to 20 cm above the table (minimum 10 cm). The observation space is detailed in Table 6.

The robot arm is controlled in raw joint velocity mode at 20 Hz. The action space is 9-dimensional as detailed in Table 5. There are no virtual walls and the robot's movement is solely restricted by the velocity limits and the objects in the scene.

Table 6: Observation used in the experiments with the Kinova Jaco Arm. An object's pose is represented as its world coordinate position and quaternion. The lid position and velocity are only used in the *Clean-Up* task. In the table, $m$ denotes meters, $rad$ denotes radians, and $q$ refers to a quaternion in arbitrary units ($au$).

| Entry | Dimensions | Unit |
|---|---|---|
| Joint Position (Arm) | 6 | rad |
| Joint Velocity (Arm) | 6 | rad/s |
| Joint Position (Hand) | 3 | rad |
| Joint Velocity (Hand) | 3 | rad/s |
| TCP Position | 3 | m |
| Touch Force (Fingers) | 3 | N |
| Red Cube Pose | 7 | m, au |
| Red Cube Velocity | 6 | m/s, dq/dt |
| Red Cube Relative Position | 3 | m |
| Blue Cube Pose | 7 | m, au |
| Blue Cube Velocity | 6 | m/s, dq/dt |
| Blue Cube Relative Position | 3 | m |
| Lid Position | 1 | rad |
| Lid Velocity | 1 | rad/s |

Analogous to *Pile1* and the SAC-X setup, we use 10 different task for *Pile2*. The first 8 tasks are seen as auxiliary tasks, that the agent uses to learn the main *two* tasks *PILE_RED* and *PILE_BLUE*, which represent stacking the red cube on the blue cube and stacking the blue cube on the red cube respectively. The tasks used in the experiment are:

- *REACH(R)* = $stol(d(TCP, R), 0.01, 0.25)$:
  Minimize the distance of the TCP to the red cube.

- *REACH(B)* = $stol(d(TCP, B), 0.01, 0.25)$:
  Minimize the distance of the TCP to the blue cube.

- *MOVE(R)* = $slin(|\,linvel(R)\,|, 0, 1)$:
  Move the red cube.

- *MOVE(B)* = $slin(|\,linvel(B)\,|, 0, 1)$:
  Move the blue cube.

- *LIFT(R)* = $btol(pos_z(R), 0.05)$
  Increase the z-coordinate of the red cube to more than 5cm relative to the table.

- *LIFT(B)* = $btol(pos_z(B), 0.05)$
  Increase the z-coordinate of the blue cube to more than 5cm relative to the table.

- *ABOVE_CLOSE(R, B)* = $above(R, B) * stol(d(R, B), 0.05, 0.2)$
  Bring the red cube to a position above of and close to the blue cube.

- *ABOVE_CLOSE(B, R)* = $above(B, R) * stol(d(R, B), 0.05, 0.2)$
  Bring the blue cube to a position above of and close to the red cube.

- *PILE(R)*:
  Place the red cube on another object (touches the top). Only given when the cube doesn't touch the robot or the table.

- *PILE(B)*:
  Place the blue cube on another object (touches the top). Only given when the cube doesn't touch the robot or the table.

The sparse reward *above(A, B)* is given by comparing the bounding boxes of the two objects *A* and *B*. If the bounding box of object A is completely above the highest point of object B's bounding box, *above(A, B)* is 1, otherwise *above(A, B)* is 0.

### A.5.3   CLEAN-UP

The *Clean-Up* task is also taken from Riedmiller et al. (2018) and builds on the setup described for the *Pile2* task. Besides the two cubes, the work-space contains an additional box with a moveable lid,

that is always closed initially (see Figure 9). The agent's goal is to clean up the scene by placing the cubes inside the box. In addition to the observations used in the *Pile2* task, the agent observes the lid's angle and it's angle velocity.

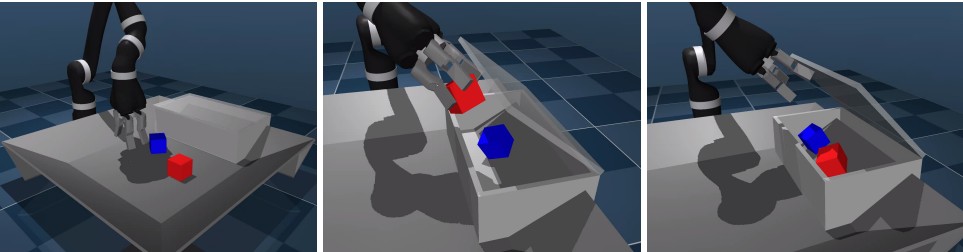

Figure 9: The *Clean-Up* task set-up in simulation. The task is solved when both bricks are in the box.

Analogous to *Pile2* and the SAC-X setup, we use 13 different task for *Clean-Up*. The first 12 tasks are seen as auxiliary tasks, that the agent uses to learn the main task *ALL_INSIDE_BOX*. The tasks used in this experiments are:

- *REACH(R)* = $stol(d(TCP, R), 0.01, 0.25)$:
  Minimize the distance of the TCP to the red cube.

- *REACH(B)* = $stol(d(TCP, B), 0.01, 0.25)$:
  Minimize the distance of the TCP to the blue cube.

- *MOVE(R)* = $slin(|linvel(R)|, 0, 1)$:
  Move the red cube.

- *MOVE(B)* = $slin(|linvel(B)|, 0, 1)$:
  Move the blue cube.

- *NO_TOUCH* = $1 - GRASP$
  Sparse binary reward, given when neither of the touch sensors is active.

- *LIFT(R)* = $btol(pos_z(R), 0.05)$
  Increase the z-coordinate of the red cube to more than 5cm relative to the table.

- *LIFT(B)* = $btol(pos_z(B), 0.05)$
  Increase the z-coordinate of the blue cube to more than 5cm relative to the table.

- *OPEN_BOX* = $slin(angle(lid), 0.01, 1.5)$
  Open the lid up to 85 degrees.

- *ABOVE_CLOSE(R, BOX)* = $above(R, BOX) * btol(|d(R, BOX)|, 0.2)$
  Bring the red cube to a position above of and close to the box.

- *ABOVE_CLOSE(B, BOX)* = $above(B, BOX) * btol(|d(B, BOX)|, 0.2)$
  Bring the blue cube to a position above of and close to the box.

- *INSIDE(R, BOX)* = $inside(R, BOX)$
  Place the red cube inside the box.

- *INSIDE(B, BOX)* = $inside(R, BOX)$
  Place the blue cube inside the box.

- *INSIDE(ALL, BOX)* = $INSIDE(R, BOX) * INSIDE(B, BOX)$
  Place the all cubes inside the box.

The sparse reward *inside(A, BOX)* is given by comparing the bounds of the object A and the box. If the bounding box of object A is completely within the box's bounds *inside(A, BOX)* is 1, otherwise *inside(A, BOX)* is 0.

## A.6 MULTITASK RESULTS

### A.6.1 PILE1 – ALL TASKS

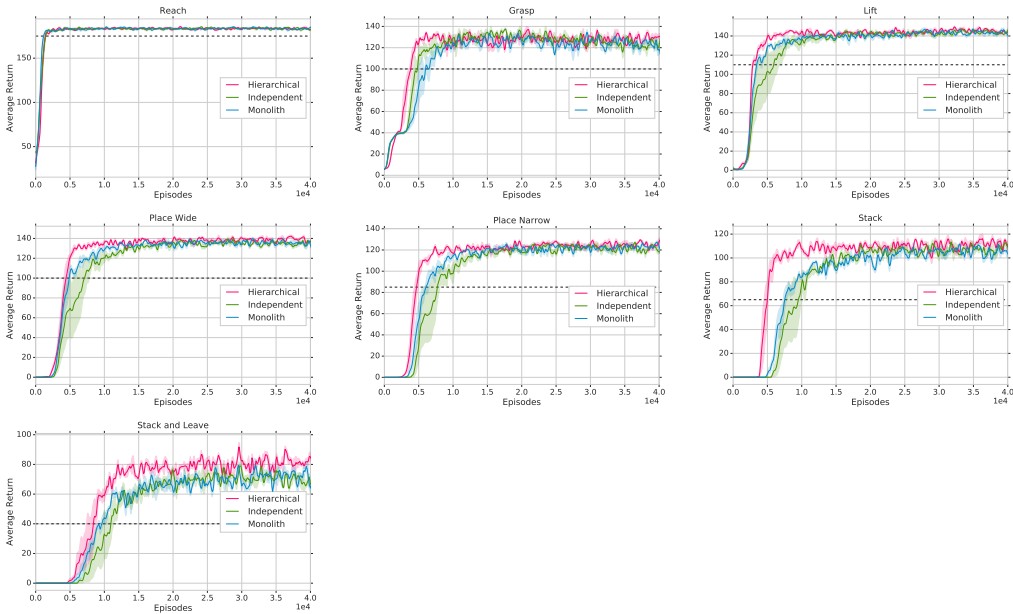

Figure 10: Complete results for all tasks from the Pile1 domain. The dotted line represents standard SAC-U after the same amount of training. Results show that using hierarchical policy leads to best performance.

### A.6.2    PILE2 – ALL TASKS

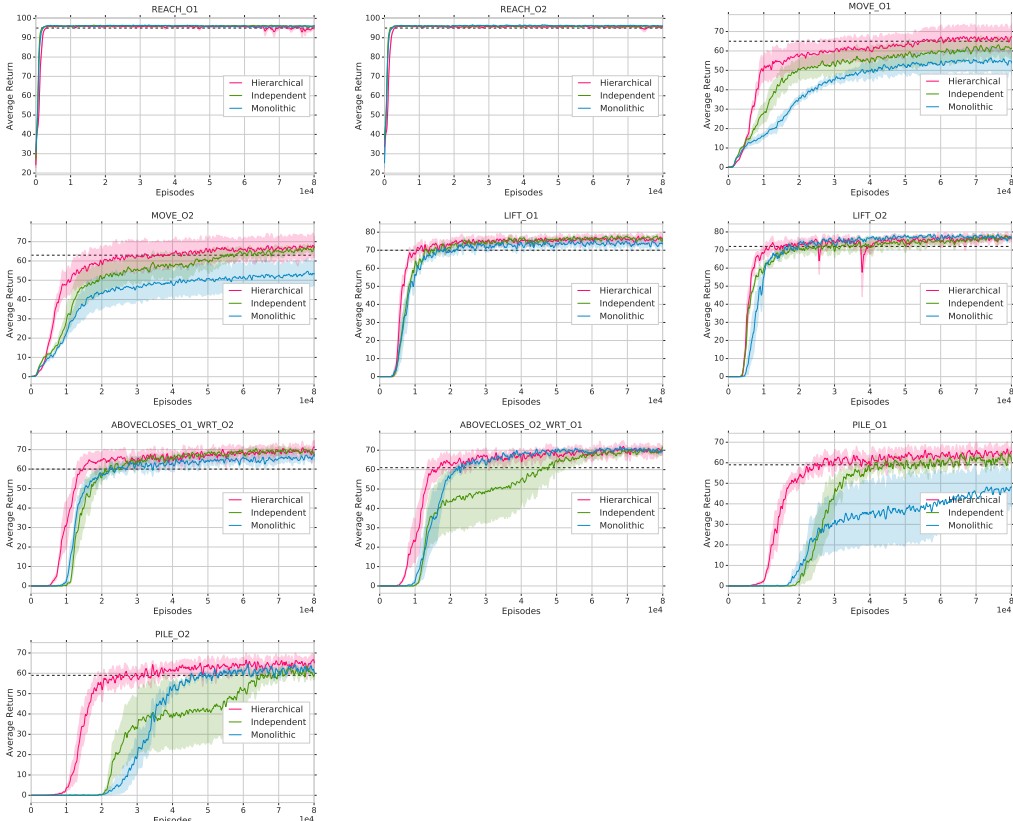

Figure 11: Complete results for all tasks from the Pile2 domain. Results show that using hierarchical policy leads to best performance. The dotted line represents standard SAC-U after the same amount of total training time.

### A.6.3 CLEANUP2 – ALL TASKS

Figure 12: Complete results for all tasks from the Cleanup2 domain. Results show that using hierarchical policy leads to best performance. The dotted line represents standard SAC-U after the same amount of total training time.

### A.7 PHYSICAL ROBOT PILE1 – ALL TASKS

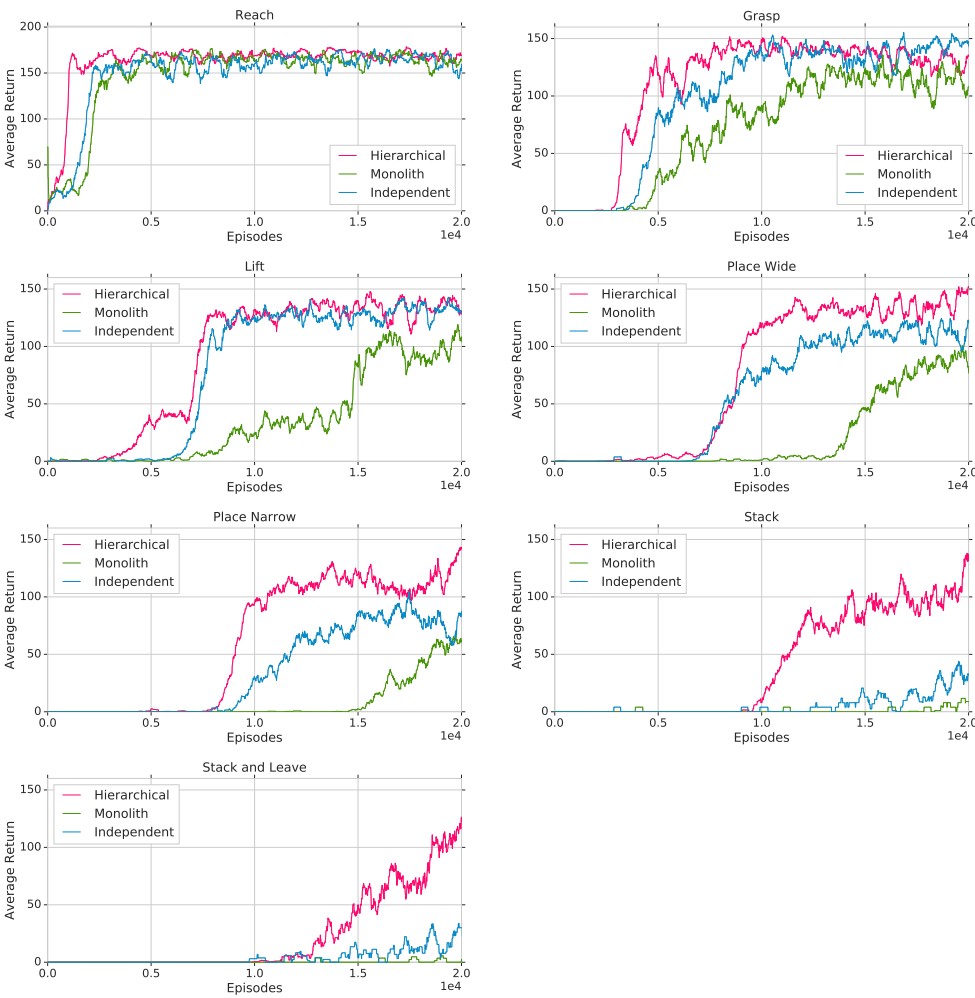

Figure 13: Complete results for all tasks on the real robot Pile1 domain. Results show that using hierarchical policy leads to best performance.

## A.8 Additional Multitask Ablations

### A.8.1 Importance of Regularization

Coordinating convergence progress in hierarchical models can be challenging but can be effectively moderated by the KL constraints. We perform an ablation study varying the strength of KL constraints on the high-level controller between prior and the current policy during training – demonstrating a range of possible degenerate behaviors.

As depicted in Figure 14, with a weak KL constraint, the high-level controller can converge too quickly leading to only a single sub-policy getting a gradient signal per step. In addition, the categorical distribution tends to change at a high rate, preventing successful convergence for the low-level policies. On the other hand, the low-level policies are missing task information to encourage decomposition as described in Section 3.2. This fact, in combination with strong KL constraints, can prevent specialization of the low-level policies as the categorical remains near static, finally leading to no or very slow convergence. As long as a reasonable constraint is picked (here a range of over 2 orders of magnitude), convergence is fast and the final policies obtain high quality for all tasks. We note that no tuning of the constraints is required across domains and the range of admissible constraints is quite broad.

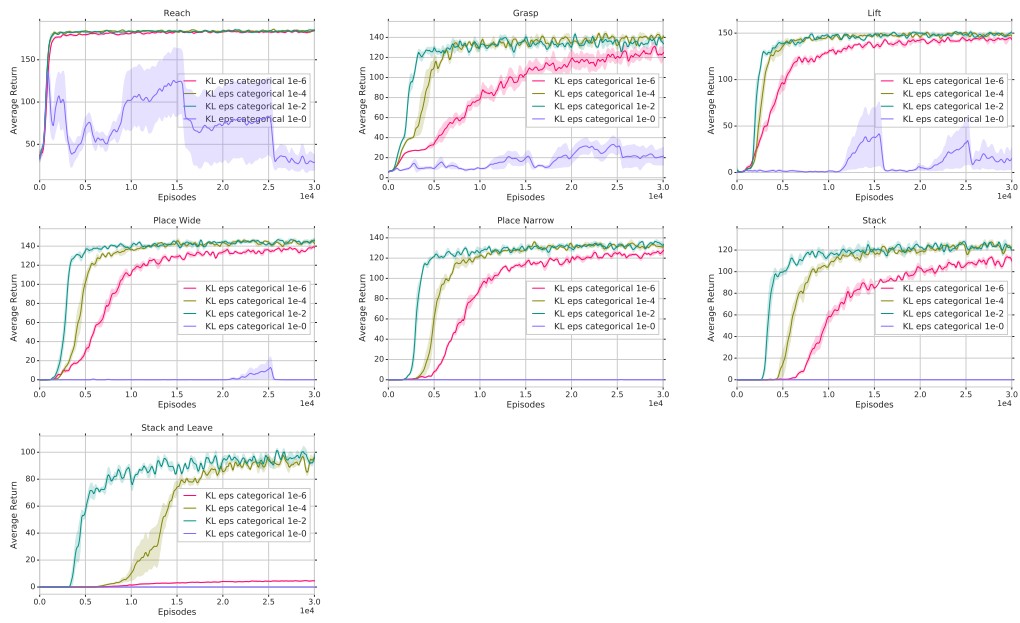

Figure 14: Complete results for sweeping the KL constraint between 1e-6 and 1. in the Pile1 domain. For very weak constraints the model does not converge successfully, while for very strong constraints it only converges very slowly.

### A.8.2 Impact of Data Rate

Evaluating in a distributed off-policy setting enables us to investigate the effect of different rates for data generation by controlling the number of actors. Figure 15 demonstrates how the different agents converge slower lower data rates (changing from 5 to 1 actor). These experiments are highly relevant for the application domain as the number of available physical robots for real-world experiments is typically highly limited. To limit computational cost, we focus on the simplest domain from Section 4.2, Pile1, in this comparison.

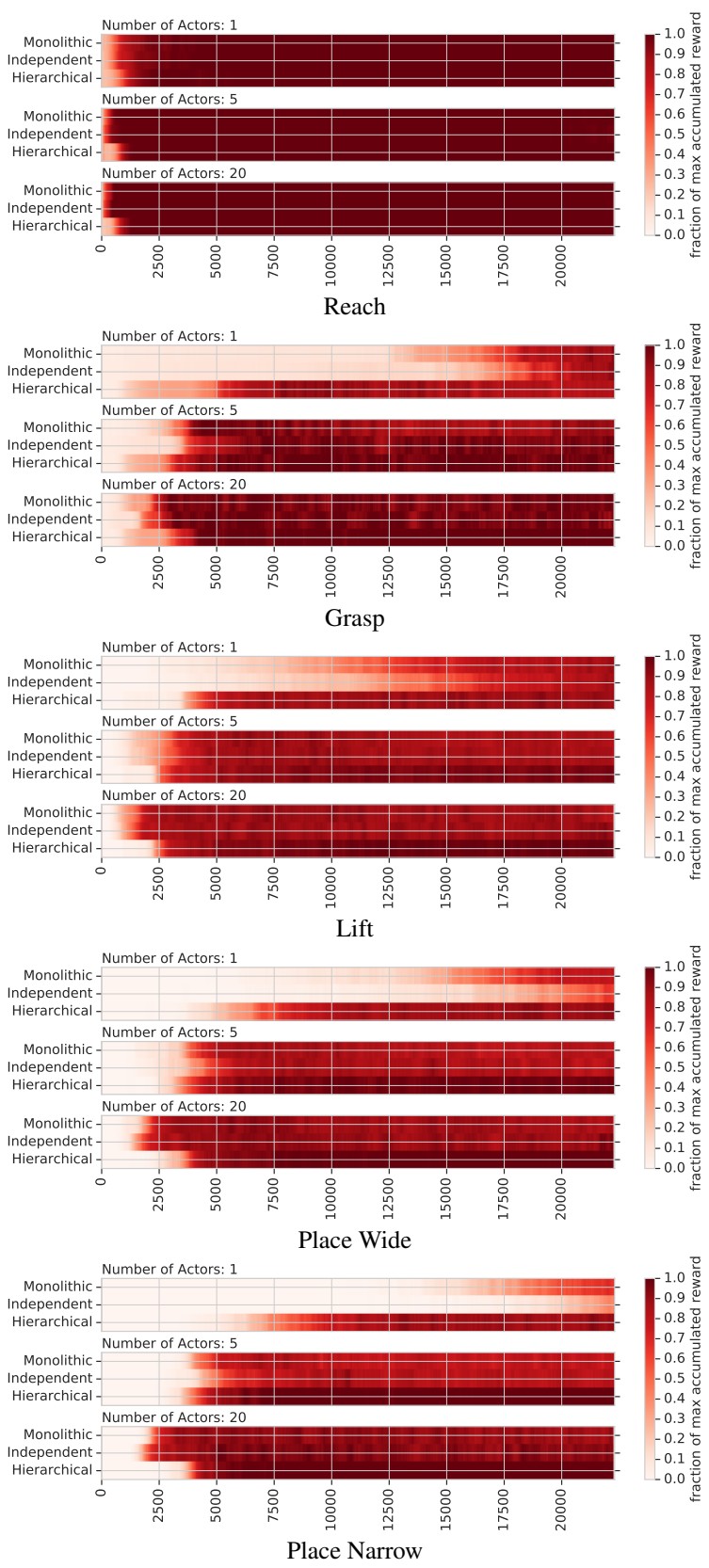

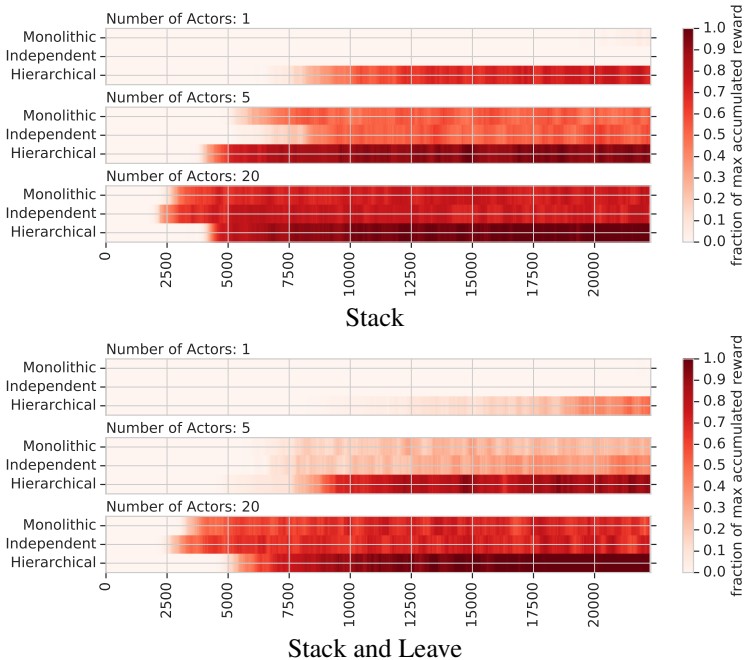

Figure 15: Complete results for ablating the number of data-generating actors in the Pile1 domain. We can see that the benefit of hierarchical policies is stronger for more complex tasks and lower data rates. However, even with 20 actors we see better final performance and stability

### A.8.3 NUMBER OF COMPONENT POLICIES

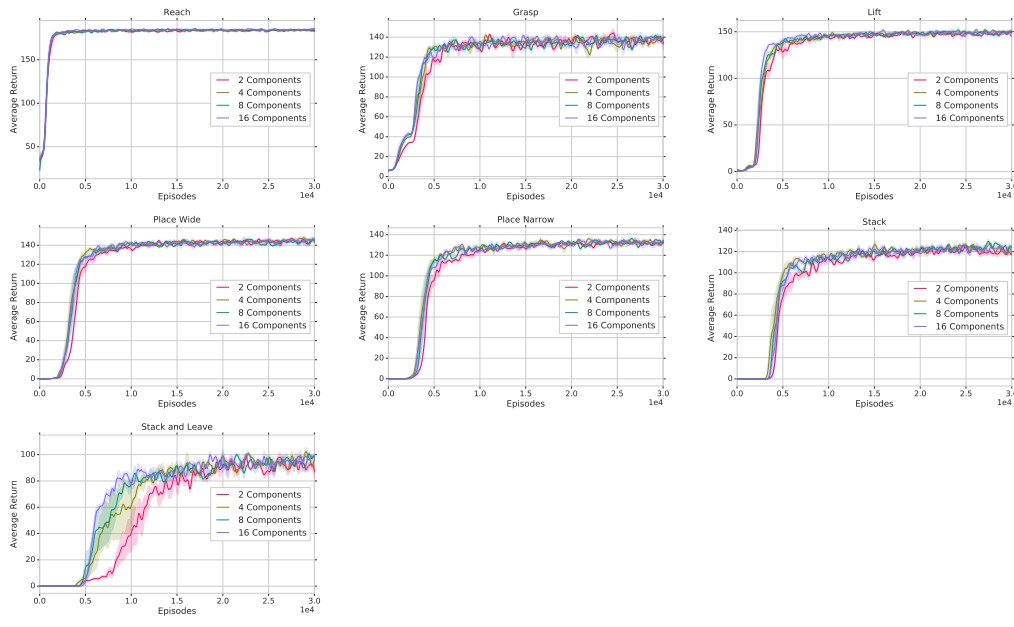

Figure 16: Complete results 2,4,8 and 16 low-level policies in the Pile1 domain. The approach is robust with respect to the number of sub-policies and we will build all further experiments on setting the components equal to the number of tasks.

### A.9 SEQUENTIAL TRANSFER EXPERIMENTS

To additionally investigate performance in adapting trained multitask policies to novel tasks, we train agents to fulfill all but the final task in the Pile1 and Cleanup2 domains and subsequently evaluate training the models on the final task. We consider two settings for the final policy by introducing only a new high-level controller (Sequential-Only-HL) or both an additional shared component as well as a new high-level controller (Sequential). Figure 17 displays that in the sequential transfer setting, starting from a policy trained on a set of related tasks results in up to 5 times more data-efficiency in terms of actor episodes on the final task than training the same policy from scratch. We observe that the final task can be solved by only reusing low-level components from previous tasks if the final task is the composition of previous tasks. This is the case for the final task in Cleanup2 which can be completed by sequencing the previously learned components and in contrast to Pile1 where the final letting go of the block after stacking is not required for earlier tasks.

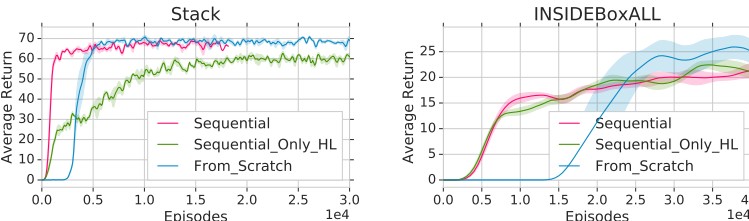

Figure 17: Sequential transfer experiments: the models are first trained with all but the final task in the Pile1 and Cleanup2 domains, and finally we train the models to adapt to the final task by either training 1- only a high-level controller or 2-a high-level controller as well as an additional component.

### A.10 HIERARCHICAL POLICIES IN REPARAMETERIZATION GRADIENT-BASED RL

To test whether the benefits of a hierarchical policy transfer to a setting where a different algorithm is used to optimize the policy we performed additional experiments using SVG (Heess et al., 2015) in place of MPO. For this purpose we use the same hierarchical policy structure as for the MPO experiments but change the categorical to an implementation that enables reparameterization with the Gumbel-Softmax trick (Maddison et al., 2016; Jang et al., 2016). We then change the entropy regularization from Equation (15) to a KL towards a target policy (as entropy regularization did not give stable learning in this setting) and use a regularizer equivalent to the distance function (per component KL's from Equation (12)) – using a multiplier of 0.05 for the regularization multiplier was found to be the best setting via a coarse grid search. This is similar to previous work on hierarchical RL with SVG (Tirumala et al., 2019).

This extension of SVG is conceptually similar to a single-step-option version of the option-critic (Bacon et al., 2017). Simplified, SVG is an off-policy actor-critic algorithm which builds on the reparametrisation instead of likelihood ratio trick (commonly leading to lower variance (Mohamed et al., 2019)). Since we do not build on temporally extended sub-policies, the algorithm simplifies to using a single critic (see Section 3.2).

The results of this experiment are depicted in Figure 18, as can be seen, for this simple domain results in mild improvements over standard SAC-U.

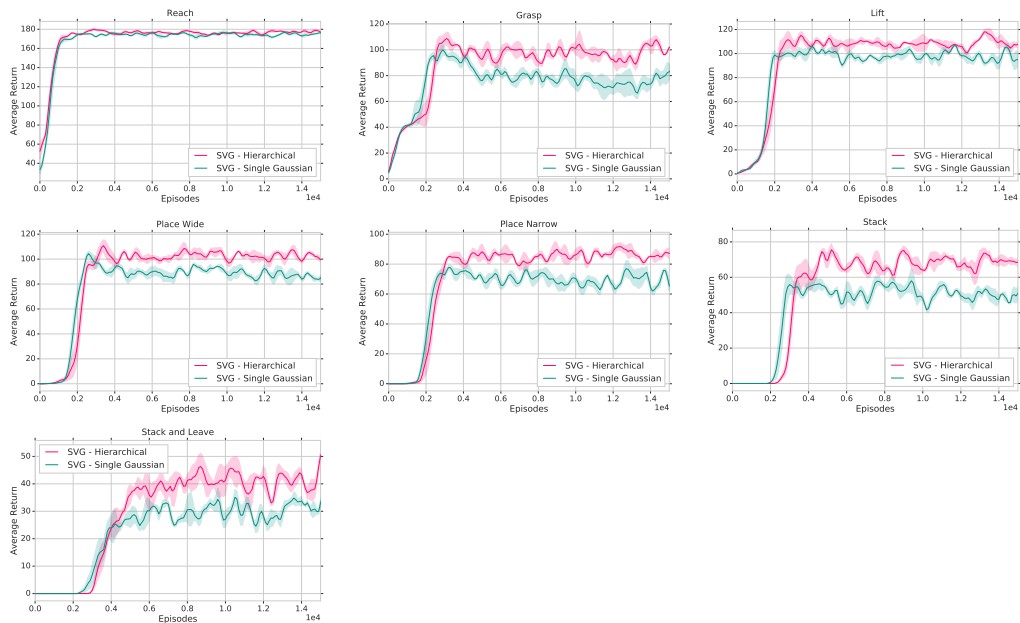

Figure 18: Complete results for evaluating SVG with and without hierarchical policy class in the Pile1 domain. Similarly to the experiments in the main paper, we can see that the hierarchical policy leads to better final performance – here for a gradient-based approach. All plots are generated by running 5 actors in parallel.

