# OpenReview forum: "Compositional Transfer in Hierarchical Reinforcement Learning"
_ICLR.cc/2020/Conference — Reject_

### Official Review · AnonReviewer2 · 2019-10-22
**Official Blind Review #2**

**Rating:** 3

**Review:**

The paper is rather interesting and is able to solve some difficult tasks. A combination of different learning techniques for acquiring structure and learning with asymmetric data are used. While the combination of methods is new I am not sure that this particular combination of methods to train an HRL policy is sufficiently novel. Is the authors can highlight the effects and contribution of how these methods are combined to indicate this better it would be good.

More detailed comments:
- In section 3.2 you mention a reference policy? Can you provide more details on this reference policy?
- IN the paper it is mentioned that the method collects data, including the reward for each task on a single state, action transition. This assumption seems rather strong. Earlier in the paper, the authors discussed the motivation for learning transferable sub-policies. In the real world, it may not be possible to collect the reward for every kind of task simultaneously.
- The first evaluation in Section 4.1 uses two humanoid environments. While these environments can be considered difficult that does not seem like the multi-task type environment the method is motivated to work well on. There is little sub-task transfer in this task.
- In section 4.1 it is noted that the version that initialized with different policy means works best. How are these means initialized?
- Is the Pile1 collection of tasks really separate tasks? It would be good to have some more details on how these are organized. There may not be a clear definition in the community what is considered a specific task but I am not overly convinced that these "different" tasks are separate. Most of them look like a similar version of pick and place.
- In Figure 2 the hierarchical method is similar in performance on the stack and leave (Pile1) set of tasks and marginally better in the Pile2 set yet does far better on the Cleanup2 set. While these are all simulations with multiple tasks is there some reasoning to why each method looks similar on the Pile1 set of tasks?
- For the robotic tasks, it is noted that again the baseline methods do well on the "Reach" task. It is shown that the RHPO does much better on the Stack task. It would be great if the authors can describe the interesting differences between the tasks. It is not clear how difficult the Stack task is and why it is largely different from Reach + Grasping.
- For the images on the right of Figure 4, It shows a comparison between the tasks and some "components" Are these components the states? The "components' are not explained well in the papers.
- There is no algorithm in the main paper which makes it a little difficult to understand the operation of the learning method. For example how exactly is the learning of the two different levels compared? It seems like they are trained together. If they are they should be compared to HIRO or HAC. How is temporal abstraction handled between the two policy layers if they are trained together?


**Experience Assessment:**

I have published in this field for several years.

**Review Assessment: Checking Correctness Of Derivations And Theory:**

I assessed the sensibility of the derivations and theory.

**Review Assessment: Checking Correctness Of Experiments:**

I carefully checked the experiments.

**Review Assessment: Thoroughness In Paper Reading:**

I read the paper at least twice and used my best judgement in assessing the paper.

---

> ### Author Response · Authors · 2019-11-11
> **Official Blind Review #2: Author Feedback**
>
> Thank you for the detailed and constructive feedback. We’re glad that the complexity of tasks, real world experiments and contributions are recognised. In the following, we aim to provide clarity to open questions in the review in particular regarding choice of methods, extensions and requirements to solve the described tasks.
>
> Our main contributions are focused on improving data-efficiency in multitask RL in complex, real-world domains. We show that the overall approach enables solving of tasks directly on a robot that go beyond what has previously been demonstrated in this setting. We focus on a simple method and show strong performance gains based on a small number of key improvements. To minimise compounding aspects, this work does not include any constraints on low level policies regarding the interface (no sub-goals inputs common to HIRO,  HAC which can be seen as orthogonal to our work) and to focus on per time-step mixture distributions to investigate composition rather than temporal abstraction.
>
> We focus on performing long-term, complex real world and simulated experiments requiring robust, data-efficient algorithms. Therefore, we develop a robust (trust region), off-policy AC algorithm, that allows data sharing across tasks (“off-policy” is used here in a strong sense since data is shared across tasks). In additional ablations, we show the importance of using a trust region constraint for the high-level controller, which has not been investigated in prior work to our knowledge. Hopefully, these algorithmic advances can additionally be transferred to other hierarchical schemes in future work.
>
> We extend two policy optimisers MPO & SVG [1] (with SVG described in Appendix 10). The extension of MPO is used for our main experiments as it is empirically more robust (see Pile1, Pile2 and Cleanup2 domains) and conceptually simpler by allowing to directly maximize probabilities under a mixture distribution in the policy improvement step without relying on MC gradient estimation via likelihood ratio or reparametrisation trick. This form of actor-critic algorithm enables us additionally to train all options and not just the executed ones based on each trajectory. Finally the generation of rewards for all tasks, common with e.g. scheduled auxiliary control or hindsight experience replay, enables to utilise the exploration capabilities for one task to benefit learning all other tasks.
>
> The extension of SVG can be seen as conceptually similar to the option-critic. Simplified, SVG is an actor-critic algorithm which builds on the reparametrisation instead of likelihood ratio trick (commonly leading to lower variance). Since we do not build on temporally extended sub-policies, we can work with a single critic thereby simplifying the algorithm. We moved some of the results into figures in the main paper to clarify this connection between both approaches.
>
> Our experiments include both complex simulated and real-world tasks and we demonstrate the data-efficiency benefits and robustness of our algorithm as well as provide further insights via an extensive set of ablations. We believe that our experiments represent a relevant step towards the deployment of (H)RL algorithms to the real world, a fundamental, open line of research.
>
>
> Please let us know if there are remaining open questions.
>
> [1] Heess, Nicolas, et al. "Learning continuous control policies by stochastic value gradients." Advances in Neural Information Processing Systems. 2015.
>
> (Minor questions are addressed in the next comment)

---

> > ### Author Response · Authors · 2019-11-11
> > **Official Blind Review #2: Author Feedback pt2**
> >
> > Minor questions (in the order of questions asked):
> >
> > - Section 3.2 reference policy: as described on the same page under the first step of policy improvement, we use the target policy as reference policy and have improved clarity when introducing the term. In our implementation, the corresponding target network is fixed for a certain number of learning steps.
> > - Rewards for all tasks simultaneously: there are clear limitations of when rewards cannot be determined for tasks in hindsight, but most commonly we work with domains where rewards simply can be computed based on state, action and observation data for a wide range of tasks. Even when we do not know about a task's existence when generating trajectories you can imagine sampling transition data later and assigning rewards for new tasks (as long as the stored data suffices for this computation). Finally, the paper includes real world tasks where exactly this is the case.
> > - Single task domains: As correctly observed , the single task domains are unable to benefit from cross-task transfer. In this context, these domains are used to investigate compositionality and show that additional incentives were required for sub-policy specialisation and the resolution performance gains via composition. Please also see section 4.1 for more details.
> > - Section 4.1 different initial means: as described we distribute the mixture’s initial means equally between minimum and maximum action range (resulting in means of -1, 0, 1 in our environments). The different initialisations provide additional incentive for the specialisation of different mixture components, which is not needed in the multitask domains. We have clarified this in the paper.
> > - Tasks in Pile1 domain: These tasks (including reaching, grasping, lifting etc) are related.  The goal of evaluating on 3 different domains (Pile1, Pile2, Cleanup2) is to show how compositional policies become more relevant for domains with more variation and less overlap between tasks. In this context Pile1 is, as correctly identified, the most similar and simple and Cleanup2 the most complex and varied domain.
> > - Improvement between domains: Similarly to the point above, we increase the complexity in multitask domains by learning to solve more tasks and less similar tasks. In this context, we were able to show that with increasing complexity (in particular regarding task similarity), compositional models become more relevant.
> > Across all environments, ‘reach’ is a comparably easy task with dense reward; as soon as the agent receives any reward, all baselines and RHPO quickly learn to solve this.  Please see page 20-24 of the appendix for all details regarding the task definitions. Intuitively, in the Pile1 domain, reaching gets a reward for getting close to the blocks, grasping for contact with the blocks, lifting for having the blocks above a specific height of the group, placing and stack depends on the positions between two blocks and stack-and-leave depends on having one brick on top of the other with the gripper further away from the stack (which is the hardest configuration). We hope this also clarifies the difference between reach+grasp and stack. For all other domains, please do have a look at the appendix of the paper.
> > - Components: we have improved clarity about this aspect throughout the paper. Components are the mixture components from the mixture of Gaussians (which is the policy distribution under RHPO).
> > - Algorithmic details: We worked hard on providing the most concise presentation of our algorithm in this paper.  Section 3.2 includes the description of policy evaluation and improvement steps which is sufficient for understanding the training procedure. Finally, the appendix should provide for all necessary aspects for reproduction of the work with the algorithm described in detail in section A.2. In this paper, we do not aim to investigate temporal abstraction and focus on training low and high level jointly by maximising the probability of actions under a mixture distribution (which consists of the high and low level policies).  However, evaluating temporal abstraction under this data-efficient and robust framework for real-world robotics tasks can be a valuable future direction.

---

> > ### Comment · AnonReviewer2 · 2019-11-13
> > **Thank you for your clarifications**
> >
> > Thank you for your clarifications
> >
> > These comments have helped clear up my understanding of some important details.

---

### Official Review · AnonReviewer3 · 2019-10-23
**Official Blind Review #3**

**Rating:** 6

**Review:**

This paper introduces a hierarchical policy structure for use in both single task and multitask reinforcement learning. The authors then assess the usefulness of such a structure in both settings on complex robotic tasks. These tasks include the stacking and reaching of blocks using a robotic hand, as an example. In addition to carrying out these experiments on simulated robots, the authors have also carried out experiments on a real Sawyer robotic arm.

The particular form of their hierarchical policy for the multitask case is as follows. The policy, which is conditioned on the current state and task index consists of a gaussian mixture, where the individual gaussian densities are conditioned on the state and a context variable. The weights of this mixture are then dependent on a density on this context variable, which is conditioned on the state and task index. The intuition behind this is that the weight portion, which is called the high level component identifies task specific information, while the low level policy learns general, shareable knowledge of the different problems.

The authors adapt the Multitask Policy Optimisation algorithm for their use by introducing an intermediate non-parametric policy, which is derived by setting KL bounds on the policy w.r.t to a reference policy. Having derived a closed-form solution to this, they go on to learn the parametric policy of interest.

The authors consider 3 settings of experiments. Firstly, they assess the benefits of the hierarchical structure for single task settings in a simulated environment. For the most part, they find that compared to a flat policy, the hierarchical structure shows benefits only if the initial means of the high-level components are sampled to be different. While the experimental results are shown to support this, further discussion of why this is the case would have been welcome.

The main benefits of the hierarchical policy are shown in the multitask case, in both simulated and real situations. In fact, the authors have shown that the hierarchical case often shows major benefits in difficult, more complicated tasks (reach vs stacking for example).

I think that the paper was very well written. It is nicely structured, with easy to read language, and without unnecessary jargon or clutter. Where necessary, the relevant extra details were provided in the Appendices.

The following are some additional notes:
1) It would have been interesting to see how the hierarchal policy faired in new tasks that were not a part of the original training set, compared to a flat multitask policy.
2) Further details about how each task is differentiated from each other in the experiments. That is, what are their different goals, which are reflected by the reward functions.

As such I recommend this paper to be weak accepted.

**Experience Assessment:**

I have read many papers in this area.

**Review Assessment: Checking Correctness Of Derivations And Theory:**

I assessed the sensibility of the derivations and theory.

**Review Assessment: Checking Correctness Of Experiments:**

I assessed the sensibility of the experiments.

**Review Assessment: Thoroughness In Paper Reading:**

I read the paper at least twice and used my best judgement in assessing the paper.

---

> ### Author Response · Authors · 2019-11-11
> **Official Blind Review #3: Author feedback**
>
> Thank you very much for the detailed review and constructive feedback. In particular, we are glad to see recognition for the paper’s clarity, the complexity of tasks and real world experiments.
>
> As requested, we extended our discussions of the results to provide further insights. In general, we also recommend the additional ablation studies in the appendix which sadly do not fit into the main paper and provide additional insight into the method.
>
> Regarding point 1)
> We are investigating transfer to new domains in Appendix 9 and are able to demonstrate significantly accelerated training on these new domains. We investigate 2 different methods for using pre-trained low level policies: one with only the pre-trained ones and one with an additional randomly initialised low level policy. If the new task is very similar to the previous domain, only the old sub-policies suffice for good performance while domains with significantly different final tasks require additional sub-policies to perform well.
>
> Regarding point 2)
> We provide a substantial description of all tasks in the appendix including quantitative description of their reward functions. To provide a better understanding of these tasks and learned solutions, we also provide videos on the paper’s website https://sites.google.com/corp/view/rhpo/
>
> Please let us know if there are remaining open questions.

---

### Official Review · AnonReviewer1 · 2019-10-23
**Official Blind Review #1**

**Rating:** 3

**Review:**

While this paper has some interesting experiments. I am quite confused about what exactly the author are claiming is the core contribution of their work. To me the proposed approach does not seem particularly novel and the idea that hierarchy can be useful for multi-task learning is also not new. While it is possible that I am missing something, I have tried going through the paper a few times and the contribution is not immediately obvious. The two improvements in section 3.2 seem quite low level and are only applicable to this particular approach to hierarchical RL. Additionally, it is very much not clear why someone, for example, would select the approach of this paper in comparison to popular paradigms like Option-Critic and Feudal Networks.

The authors mention that Feudal approaches "employ different rewards for different levels of the hierarchy rather than optimizing a single objective for the entire model as we do." Why reward decomposition at the lower levels is a problem instead of a feature isn't totally clear, but this criticism does not apply to Option-Critic models. For Option-Critic models the authors claim that "Rather than the additional inductive bias of temporal abstraction, we focus on the investigation of composition as type of hierarchy in the context of single and multitask learning while demonstrating
the strength of hierarchical composition to lie in domains with strong variation in the objectives such as in multitask domains." First of all, I should point out that [1] looked at applying Option-Critic in a many task setting and found both that there was an advantage to hierarchy and an advantage to added depth of hierarchy. Additionally, it is well known that Option-Critic approaches (when unregularized) tend to learn options that terminate every step [2].  So, if you generically apply Option-Critic, it would in fact be possible to disentangle the inductive bias of hierarchy from the inductive bias of temporal abstraction by using options that always terminate.

In comparison to past frameworks, the approach of this paper seems less theoretically motivated. It certainly does not seem justified to me to just assume this framework and disregard past successful approaches even as a comparison. While the experiments show the value of hierarchy, they do not show the value of this particular method of creating hierarchy. The feeling I get is that the authors are trying to make their experiments less about what they are proposing in this paper and more about empirical insights about the nature of hierarchy overall. If this is the case, I feel like the empirical results are not novel enough to create value for the community and too tied to a particular approach to hierarchy which does not align with much of the past work on HRL.

[1] "Learning Abstract Options". Matthew Riemer, Miao Liu, and Gerald Tesauro. NeurIPS-18.
[2] "When Waiting is not an Option: Learning Options with a Deliberation Cost" Jean Harb, Pierre-Luc Bacon, Martin Klissarov, and Doina Precup. AAAI-18.

**Experience Assessment:**

I have published one or two papers in this area.

**Review Assessment: Checking Correctness Of Derivations And Theory:**

N/A

**Review Assessment: Checking Correctness Of Experiments:**

I assessed the sensibility of the experiments.

**Review Assessment: Thoroughness In Paper Reading:**

I read the paper at least twice and used my best judgement in assessing the paper.

---

> ### Author Response · Authors · 2019-11-11
> **Official Blind Review #1: Author feedback**
>
> Thank you very much for the detailed feedback. We’re glad that the complexity of tasks and real world experiments are recognised and worked to address open questions and clarify contributions and goal of the paper in the following sections.
>
> Our main contributions are focused on improving data-efficiency in multitask RL in complex, real-world domains. We show that the overall approach enables solving of tasks directly on a robot that go far beyond what has previously been demonstrated in this setting. First, we extend existing investigations into hierarchical RL with a focus on robustness (see points below regarding trust-region) and benefits and challenges for compositionality (see single and multi task).
>
> We focus on a simple method and show strong performance gains based on a small number of key improvements. To minimise compounding aspects, this work does not include any constraints on low level policies regarding the interface (no sub-goals inputs common to HIRO, FuN and HAC which can be seen as orthogonal to our work) and to focus on per time-step mixture distributions to investigate composition rather than temporal abstraction. Here, we do not claim that 1-step options, or the lack of intrinsic rewards are our contribution but that the conceptually simple setup is good enough to achieve dramatic improvements in data efficiency, without the other ingredients.
>
> We focus on performing complex, long-term real world and simulated experiments requiring robust, data-efficient algorithms. Therefore, we develop a robust (trust region), off-policy AC algorithm, that allows data sharing across tasks (“off-policy” is used here in a strong sense since data is shared across tasks). This setting (both regarding off-policy and trust region improvement) is rather different from the typical near on-policy settings studied often in the literature and in particularly differs from typical application domains for the mentioned work on OC and FuN. In additional ablations, we show the importance of using a trust region constraint for the high-level controller, which has not been investigated in prior work to our knowledge. Hopefully, these algorithmic advances can additionally be transferred to other hierarchical schemes in future work.
>
> We extend two policy optimisers MPO & SVG[1] (with SVG described in Appendix 10). The extension of MPO is used for our main experiments as it is empirically more robust (see Pile1, Pile2 and Cleanup2 domains) and conceptually simpler by allowing to directly maximize probabilities under a mixture distribution in the policy improvement step without relying on MC gradient estimation via likelihood ratio or reparametrisation trick. This form of actor-critic algorithm enables us additionally to train all options and not just the executed ones based on each trajectory. Finally the generation of rewards for all tasks, common with e.g. scheduled auxiliary control or hindsight experience replay, enables to utilise the exploration capabilities for one task to benefit learning all other tasks.
>
> We have improved the submission to better point out connections to prior work by moving additional results into figures in the main paper to clarify connections. Our extension of SVG [1] trains mixture of Gaussians policies by using the Gumbel Softmax trick (in the Appendix). This leads to increased performance compared to flat Gaussian policies and actually is conceptually similar to the option-critic. Simplified, SVG is an actor-critic algorithm which builds on the reparametrisation instead of likelihood ratio trick (commonly leading to lower variance). Since we do not build on temporally extended sub-policies we can work with a single critic, thereby simplifying the algorithm.
>
> The idea of using hierarchy for multitask (reinforcement) learning is indeed not new and has not been new for many years. However there has been process towards improved data efficiency and better understanding of algorithms and mechanisms (for example in the mentioned related work). Our experiments include both complex simulated and real-world tasks and we demonstrate the data-efficiency benefits and robustness of our algorithm as well as provide further insights via an extensive set of ablations. We believe that our experiments represent a relevant step towards the deployment of RL algorithms to the real world, a fundamental, open line of research.
>
> Finally, we are of course happy to extend our literature review by the suggested references.
> Please let us know if there are remaining open questions.
>
> [1] Heess, Nicolas, et al. "Learning continuous control policies by stochastic value gradients." Advances in Neural Information Processing Systems. 2015.

---

### Decision · Program_Chairs · 2019-12-19

**Decision:**

Reject

**Comment:**

This paper is concerned with improving data-efficiency in multitask reinforcement learning problems. This is achieved by taking a hierarchical approach, and learning commonalities across tasks for reuse. The authors present an off-policy actor-critic algorithm to learn and reuse these hierarchical policies.

This is an interesting and promising paper, particularly with the ability to work with robots. The reviewers did however note issues with the novelty and making the contributions clear. Additionally, it was felt that the results proved the benefits of hierarchy rather than this approach, and that further comparisons to other approaches are required. As such, this paper is a weak reject at this point.